# Conversational Recommender System and Large Language Model Are Made for Each Other in E-commerce Pre-sales Dialogue

Yuanxing Liu[1]    Wei-Nan Zhang[1†]    Yifan Chen[1]    Yuchi Zhang[1]    Haopeng Bai[1]

Fan Feng[2]    Hengbin Cui[2]    Yongbin Li[2]    Wanxiang Che[1]

[1]Research Center for Social Computing and Information Retrieval
Harbin Institute of Technology, China
[2]Independent, China

{yxliu, wnzhang, yfchen, yczhang, hpbai, car}@ir.hit.edu.cn

{fengfan.blender, alexcui.chb, liyb821}@gmail.com

## Abstract

E-commerce pre-sales dialogue aims to understand and elicit user needs and preferences for the items they are seeking so as to provide appropriate recommendations. Conversational recommender systems (CRSs) learn user representation and provide accurate recommendations based on dialogue context, but rely on external knowledge. Large language models (LLMs) generate responses that mimic pre-sales dialogues after fine-tuning, but lack domain-specific knowledge for accurate recommendations. Intuitively, the strengths of LLM and CRS in E-commerce pre-sales dialogues are complementary, yet no previous work has explored this. This paper investigates the effectiveness of combining LLM and CRS in E-commerce pre-sales dialogues, proposing two collaboration methods: *CRS assisting LLM* and *LLM assisting CRS*. We conduct extensive experiments on a real-world dataset of E-commerce pre-sales dialogues. We analyze the impact of two collaborative approaches with two CRSs and two LLMs on four tasks of E-commerce pre-sales dialogue. We find that collaborations between CRS and LLM can be very effective in some cases.

## 1 Introduction

E-commerce pre-sales dialogue refers to a dialogue between a user and a customer service staff before the purchase action (Chen et al., 2020; Zhao et al., 2021). A high-quality pre-sales dialogue can greatly increase the purchase rate of a user. However, there are many challenges to providing a high-quality pre-sales dialogue service (Liu et al., 2023b). Refine. Figure 1 shows an example of an e-commerce pre-sales dialogue. The bot needs to interact with the user, understanding the user's needs and responding with understandable words. Additionally, it should offer appropriate recommendations and elicit further preferences from the user.

---

†Corresponding author.

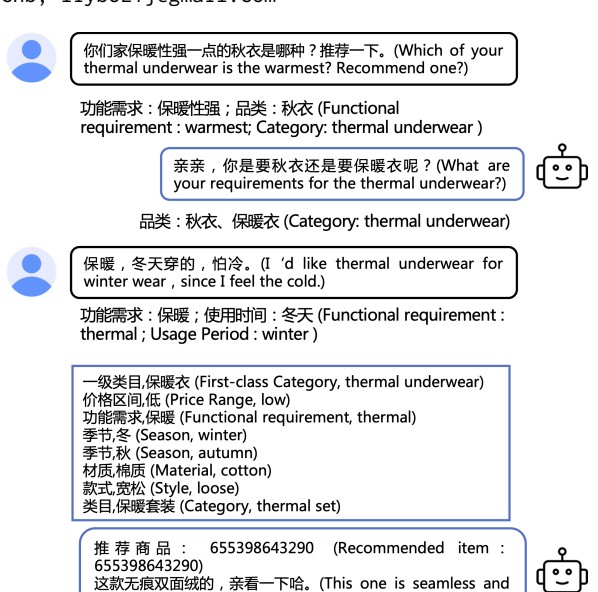

Figure 1: Example of E-commerce pre-sales dialogue.

Conversational recommender systems (CRSs) aim to learn relationships between user preferences and candidate product representations to provide accurate recommendations (Li et al., 2018) and generate responses related to recommended products (Liang et al., 2021). However, understanding user preferences from dialogues relies heavily on external knowledge (Chen et al., 2019), e.g. DBpedia and ConceptNet. With an external knowledge base, CRS is able to recognize the entities present in the context, but it still has difficulty understanding the semantic information within the context.

Large language models (LLMs), which have numerous parameters pre-trained on a large amount of data, possess a wealth of knowledge that enables people to interact with them using natural language (Brown et al., 2020; Ouyang et al., 2022). With supervised fine-tuning tasks, LLMs can handle diverse user needs descriptions in pre-sales dialogues. However, LLMs lack information about candidate products, which makes them not suitable for providing domain-specific recommendations.

What will happen when LLMs and CRSs collaborate? In this paper, we explore two types of collaborations between LLMs and CRSs in E-commerce pre-sales dialogues: (i) *LLM assisting CRS*. (ii) *CRS assisting LLM*. When a LLM assists a CRS, we append the generated response of the LLM to the input of the CRS. For the recommendation task, we incorporate the representation of the product predicted by the LLM into the calculation of the user representation. When a CRS assists a LLM, we append the predictions of the CRS to the input of the LLM. For the recommendation task, we insert the recommendation list, while the other tasks insert the text.

Specifically, we explore the effectiveness of collaborations on a real-world dataset of E-commerce pre-sales dialogues, namely U-NEED (Liu et al., 2023b). U-NEED contains pre-sales dialogues in five top categories and supports four key tasks in E-commerce pre-sales dialogue: (i) pre-sales dialogue understanding (ii) user needs elicitation (iii) user needs-based recommendation and (iv) pre-sales dialogue generation. We select two popular open source LLMs, ChatGLM-6B and Chinese-Alpaca-7B, as well as two latest CRSs, Bart-based CRS and CPT-based CRS. We report experimental results for each combination of collaborations on the four challenging tasks. Experimental results demonstrate that the collaboration between LLM and CRS is effective on three tasks: pre-sales dialogue understanding, user needs elicitation and user needs-based recommendation.

Main contributions of this paper are as follows:

- To the best of our knowledge, we are the first to explore collaboration between LLM and CRS in a real-world scenario, namely E-commerce pre-sales dialogue.

- We propose methods for two types of collaborations between LLMs and CRSs, i.e., *CRS assisting LLM* and *LLM assisting CRS*.

- Extensive experimental results on a real-world E-commerce pre-sales dialogue dataset indicate the effectiveness and potential of collaborations between LLMs and CRSs.

## 2 Related Work

We review the related work along two lines: (i) conversational recommendation and (ii) large language models (LLMs) for recommendation.

### 2.1 Conversational recommendation

Conversational recommender systems (CRSs) aim to provide real-time recommendations based on users' dynamic preferences through natural language interactions (Gao et al., 2021; Jannach et al., 2021). Early work focus on: (i) question-based user preference elicitation (Zou et al., 2020; Hu et al., 2022a), (ii) multi-turn conversational recommendation strategies (Lei et al., 2020a,b), (iii) exploration–exploitation trade-offs (Fu et al., 2021; Wong et al., 2021; Zhang et al., 2020), (iv) user preference modeling with external knowledge (Zhou et al., 2022; Chen et al., 2019; Ma et al., 2021; Zhou et al., 2021; Ren et al., 2022), (v) dialogue strategies (Liu et al., 2020; Zhou et al., 2020; Hayati et al., 2020) and (vi) generating persuasive responses (Liang et al., 2021). Recently, some work (Deng et al., 2023; Wang et al., 2022a,b) utilize pre-trained language models (PLMs) as the foundation to build unified CRSs, capable of performing various tasks using a single model, instead of multiple components.

The emergence of LLMs has undoubtedly impacted CRS-related researches. However, previous work barely explores the collaboration between conversational language models and CRSs on tasks related to conversational recommendation. We investigate the collaboration between LLM and CRS in E-commerce pre-sales dialogues.

### 2.2 LLMs for recommendation

Large language models (LLMs), such as GPT-3 (Brown et al., 2020), InstructGPT (Ouyang et al., 2022), PaLM (Chowdhery et al., 2022), Bloom (Scao et al., 2022), LLaMA (Touvron et al., 2023) and GLM (Du et al., 2022), have gained attention for their natural language understanding and generation capabilities (Zhao et al., 2023). Recent studies have examined the performance of ChatGPT (OpenAI, 2022) in tasks such as passage re-ranking (Sun et al., 2023) and recommendation (Wang et al., 2023; Liu et al., 2023a). ChatGPT has also been applied to domains like augmenting recommender systems (Gao et al., 2023). Additionally, Friedman et al. (2023) propose a roadmap for utilizing LLM to build a controllable and explainable CRS for YouTube videos.

Different from previous work using LLM to enhance CRS, we systematically investigate the effectiveness of combining LLM and CRS, i.e., *LLM assisting CRS* and *CRS assisting LLM*, which provides insights for future research on CRSs.

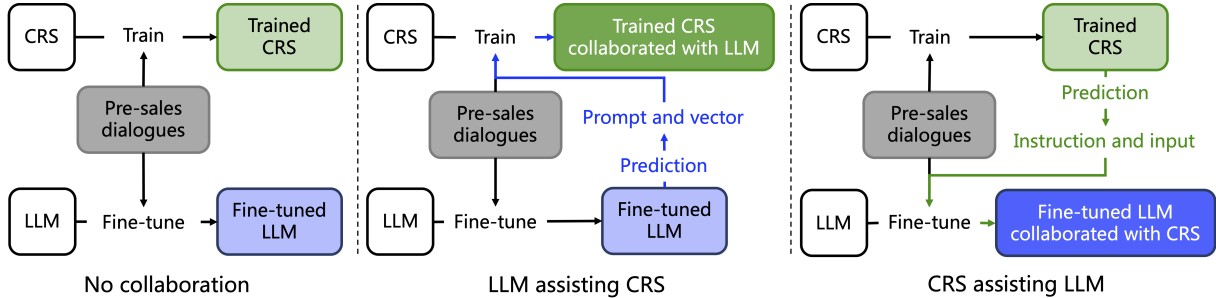

Figure 2: A comparison of the three types of collaboration between a CRS and a LLM. We explore the collaboration between LLM and CRS, i.e., *LLM assisting CRS* and *CRS assisting LLM*, and we compare the three in detail in §5.

## 3 Method

### 3.1 Overview

In this paper, we explore the collaboration of a conversational recommender system (CRS) and a large language model (LLM). Figure 2 provides an illustration of our collaboration framework.

**LLM assisting CRS.** We leverage the prediction results of a LLM to support a CRS. Initially, we fine-tune a LLM using pre-sales dialogues, following the method described in Section 3.2. Subsequently, we incorporate the prediction results of the fine-tuned large model into the training process of the CRS, via prompts and vectors. For a detailed description, refer to Section 3.5.

**CRS assisting LLM.** We utilize the prediction results of a CRS to assist a LLM. Initially, we train a CRS using pre-sales dialogues, following the approach outlined in Section 3.3. Subsequently, we integrate the prediction results of the trained CRS into the instructions and inputs to optimize the process of fine-tuning the LLM. For further details, see Section 3.4.

In this paper, we explore the effectiveness of collaboration by analyzing the impact of collaboration between CRS and LLM on the performance of four tasks in E-commerce pre-sales dialogue (Liu et al., 2023b). These tasks include: (i) pre-sales dialogue understanding (ii) user needs elicitation (iii) user needs-based recommendation and (iv) pre-sales dialogue generation. Due to space constraints, we provide detailed definitions of these tasks in Appendix A.

### 3.2 LLMs for E-commerce pre-sales dialogue

We introduce the method of fine-tuning a large language model (LLM) using pre-sales dialogues.

**Instruction data.** Each sample within the training, validation, and test sets consists of "instruction", "input" and "output". The "instruction" com-

prises several sentences that introduce the task's objective. The "input" contains the necessary information for completing the task. For instance, in the case of a user needs-based recommendation task, the "input" encompasses the user's needs, candidate products, and related product knowledge. The "output" remains consistent with the original task. Figure 3 shows an example of the instruction data corresponding to the user needs elicitation task. Additional examples of various tasks can be found in Appendix F. Note that the original user needs-based recommendation task involves numerous candidates, with each product possessing extensive attribute knowledge, the "input" surpasses the maximum input length permitted by LLMs. Consequently, in practice, we limit the number of candidates to 20.

**Base LLMs and fine-tuning.** We select ChatGLM and Chinese-Alpaca-7B as base LLMs due to their openness and commendable performance in Chinese basic semantic understanding. ChatGLM is an open bilingual language model built upon General Language Model (GLM) framework (Zeng et al., 2023), with 6.2 billion parameters.[1] LLaMA (Touvron et al., 2023) is a decoder-only, foundational large language model based on the transformer architecture (Vaswani et al., 2017). The Chinese LLaMA model is an extension of the original LLaMA model, incorporating an expanded Chinese vocabulary and undergoing secondary pre-training using Chinese data (Cui et al., 2023). We adopt the Chinese Alpaca model, which builds upon the aforementioned Chinese LLaMA model by incorporating instruction data for fine-tuning.[2] To carry out the fine-tuning process, we follow the official method provided by ChatGLM6B/Chinese-Alpaca-Plus-7B, using LoRA (Hu et al., 2022b).

[1] https://github.com/THUDM/ChatGLM
[2] https://github.com/ymcui/Chinese-LLaMA-Alpaca

## 3.3 CRSs for E-commerce pre-sales dialogue

We introduce the method of train a conversational recommender system (CRS) for pre-sales dialogues. We adopt UniMIND (Deng et al., 2023) as our base CRS, as it focus on multiple tasks in conversational recommendation. The recommendation candidates for UniMIND are movies. The movie title can be generated based on the prompt. However, in E-commerce pre-sales dialogue, the recommendation candidate is the product ID, which is difficult to be decoded directly. Therefore, for the user needs-based recommendation task we follow a traditional user representation-based approach (Kang and McAuley, 2018).

**Prompts.** Following Deng et al. (2023), we define the inputs and outputs of the four tasks using a unified sequence-to-sequence paradigm. We use five special tokens to indicate information segments: (i) *[user]* indicates the utterance of the user. (ii) *[system]* indicates the response from customer service staff. (iii) *[understand]* indicates the needs contained in the user utterance, i.e., attributes and attribute values. (iv) *[elicit]* indicates the attributes that the customer service staff plans to ask about user preferences. (v) *[recommend]* indicates the items that have been recommended by customer service staff. For instance, the original input $X$ can be represented as follows:

$$X_U = \text{[user]}\ u_1\ \text{[understand]}\ d_1\ \text{[system]}\ s_1\ \text{[understand]}$$
$$d_2\ \ldots\ \text{[user]}\ u_i$$
$$X_S = \text{[user]}\ u_1\ \text{[understand]}\ d_1\ \text{[system]}\ s_1\ \text{[understand]}$$
$$d_2\ \ldots\ \text{[system]}\ s_i$$
$$X_A = \text{[user]}\ u_1\ \text{[understand]}\ d_1\ \text{[elicit]}\ a_1\ \text{[system]}\ s_1$$
$$\text{[understand]}\ d_2\ \ldots\ \text{[user]}\ u_i$$
$$X_R = \text{[user]}\ u_1\ \text{[understand]}\ d_1\ \text{[recommend]}\ e_1$$
$$\text{[system]}\ s_1\ \text{[understand]}\ d_2\ \ldots\ \text{[user]}\ u_i$$
$$X_G = \text{[user]}\ u_1\ \text{[understand]}\ d_1\ \text{[system]}\ s_1\ \text{[understand]}$$
$$d_2\ \ldots\ \text{[user]}\ u_i\ \text{[elicit]}\ a_1\ \text{[recommend]}\ e_1$$

where $u_i$ is the $i$-th utterance of the user, $s_i$ is the $i$-th response of customer service staff, $d_i$ is the $i$-th user needs, $a_i$ is the $i$-th attribute to be asked, and $e_i$ is the $i$-th recommended product. We adopt natural language prompt (Raffel et al., 2020) to indicate each task:

$$Z_U = \text{``Identify attributes and values:''}$$
$$Z_A = \text{``Select an attribute to ask:''}$$
$$Z_G = \text{``Generate a response:''}$$

**Loss functions.** Following UniMIND (Deng et al., 2023), we design a unified CRS with prompts learning and multitask learning for pre-sales dialogues.

$$\mathcal{L}_\theta = \mathbb{E}_{(X,Y,Z)\sim\mathcal{D}} \sum_{l=1}^{L} \log p_{\theta(y_l|y_{<l},X,Z)}, \quad (1)$$

where $\mathcal{D} = \{\mathcal{D}_U, \mathcal{D}_A, \mathcal{D}_G\}$ denote the train-set for three tasks: pre-sales dialogue understanding, user needs elicitation and pre-sales dialogue generation. And $L$ is the length of the generated sequence. For user needs-based recommendation task, the recommendation probability and loss function are defined as follows:

$$r_i = \mathbf{CLS}(\boldsymbol{e_i}, \mathbf{Enc}(X_R)) \quad (2)$$
$$\mathcal{L}_R = -\sum_{i=1}^{|E|} e_i \log r_i, \quad (3)$$

where $\boldsymbol{e_i}$ is the trainable item embedding, $E$ is collection of candidate products and $r_i$ is recommendation probability. $\mathbf{CLS}(\cdot)$ is a classifier, we apply linear layer in practice. And $\mathbf{Enc}(\cdot)$ is the encoder, we adopt two versions: BART (Lewis et al., 2020) and CPT (Shao et al., 2021).

**Training process.** We train a CRS in two stages. In the first stage we train a CRS only on the user needs-based recommendation task, i.e., $\mathcal{L} = \mathcal{L}_R$. Since BART and CPT do not have pre-sales conversation knowledge, this step aims to warm up CRS. In the second stage we continue to train a CRS on all four tasks, i.e., $\mathcal{L} = \mathcal{L}_R + \mathcal{L}_\theta$.

## 3.4 Collaboration 1: CRS assisting LLM

We introduce the method of CRS assisting LLM. Initially, we train a CRS model following the approach outlined in Section 3.3. Subsequently, we enrich the original instructions and inputs of a LLM by incorporating the prediction outcomes from the CRS model. Finally, we fine-tune the LLM, as explained in § 3.3, utilizing the augmented instructions and inputs.

**Enhanced instruction and input.** We convert the output of a trained CRS into text and incorporate it into the input of LLM. We add additional instruction, i.e., completing the task requires considering the results of the CRS. An example is shown in the upper right corner of Figure 3. Note that in the user needs-based recommendation task, CRS can output a list of recommendations along with corresponding recommendation scores. We rank the

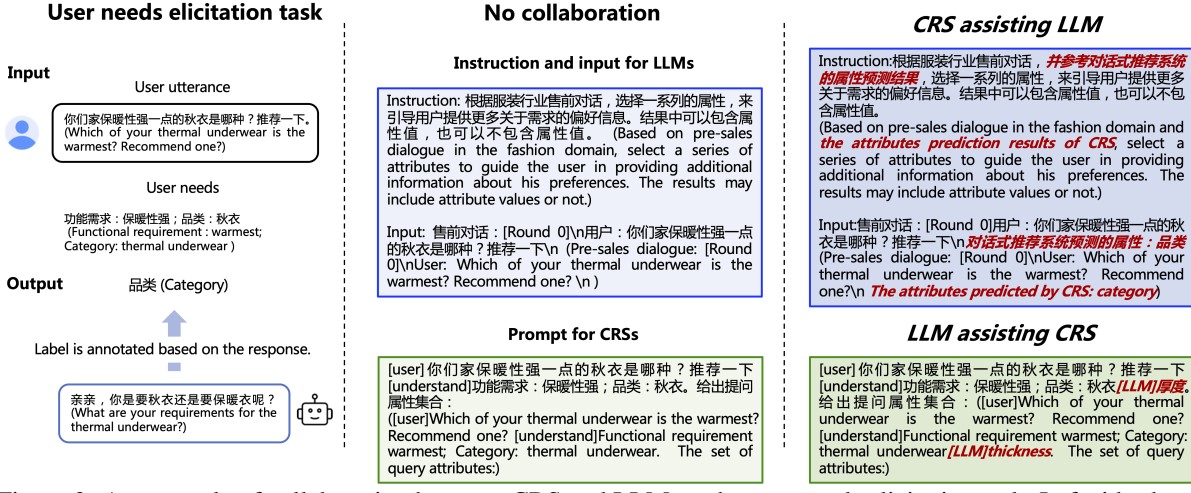

Figure 3: An example of collaboration between CRS and LLM on the user needs elicitation task. Left side shows the input and output of the task. The middle displays data used to fine-tune a LLM and train a CRS independently. The right side shows two cases of combining the two. Collaboration content is highlighted in red italics.

candidates according to their scores and include the ranking to the input of LLM. For other tasks, we only consider the final output of the trained CRS.

## 3.5 Collaboration 2: LLM assisting CRS

We introduce the method of LLM assisting CRS. Initially, we fine-tune a LLM model using the technique described in Section 3.2. Subsequently, we employ the prediction outcomes from LLM to enhance the prompt and user representation in the CRS model. Finally, we train a CRS model as outlined in § 3.3.

**Enhanced prompts.** We use the prediction results of LLM to enhance the prompts of CRS.

$$X'_U = [\text{user}] \ u_1 \ [\text{understand}] \ d_1 \ [\text{system}] \ s_1 \ [\text{understand}]$$
$$d_2 \dots [\text{user}] \ u_i \ [\text{LLM}] \ \hat{a}_i, \hat{v}_i$$
$$X'_S = [\text{user}] \ u_1 \ [\text{understand}] \ d_1 \ [\text{system}] \ s_1 \ [\text{understand}]$$
$$d_2 \dots [\text{system}] \ s_i \ [\text{LLM}] \ \hat{a}_i, \hat{v}_i$$
$$X'_A = [\text{user}] \ u_1 \ [\text{understand}] \ d_1 \ [\text{elicit}] \ a_1 \ [\text{system}] \ s_1$$
$$[\text{understand}] \ d_2 \dots [\text{user}] \ u_i \ [\text{LLM}] \ \hat{a}_i$$
$$X'_G = [\text{user}] \ u_1 \ [\text{understand}] \ d_1 \ [\text{system}] \ s_1 \ [\text{understand}]$$
$$d_2 \dots [\text{user}] \ u_i \ [\text{elicit}] \ a_1 \ [\text{recommend}] \ e_1 \ [\text{LLM}] \ \hat{s}_i$$

For $X'_U$ and $X'_S$, $\hat{a}_i$ and $\hat{v}_i$ are attributes and values involved in utterance identified by LLM for the $i$-th turn. For $X'_A$, $\hat{a}_i$ is the attribute in user needs elicitation for the $i$-th turn. For $X'_G$, $\hat{s}_i$ is the response for the pre-sales dialogue generation task generated by LLM.

**Enhanced representation.** For recommendation task, we consider the embedding of the product recommended by the fine-tuned LLM:

$$r'_i = \mathbf{CLS}(e_i, \mathbf{Enc}(X_R), \hat{e}_i) \quad (4)$$
$$\mathcal{L}'_R = -\sum_{i=1}^{|E|} e_i \log r'_i, \quad (5)$$

where $\hat{e}_i$ is the embedding of the product recommended by the fine-tuned LLM.

## 4 Experimental Settings

### 4.1 Research questions

To guide the remaining part of this paper, we set up two research questions:

- Are LLM and CRS complementary? Does the combination of LLM and CRS improve performance?

- How does the combination of CRS and LLM perform on different tasks and different categories? What are the differences between different collaboration methods?

In the Results and Analysis section, we systematically examine the outcomes of each task to answer the aforementioned two research questions.

### 4.2 Dataset

We conduct experiments on U-NEED (Liu et al., 2023b). U-NEED consists of 7,698 fine-grained annotated pre-sales dialogues, which consist of 1662, 1513, 1135, 1748, and 1640 dialogues in *Beauty*, *Phones*, *Fashion*, *Shoes* and *Electronics* categories respectively. We follow the partition of the training set, validation set and test set proposed in U-NEED.

Table 1: Performance of baseline methods on pre-sales dialogue understanding task in 3 typical categories: *Beauty*, *Fashion* and *Shoes*. Baseline results marked with * are taken from U-NEED (Liu et al., 2023b). CLLM is short for ChatGLM and ALLM is short for Chinese-Alpaca. BCRS is short for UniMIND(BART) and CCRS is short for UniMIND(CPT). The best results are highlighted in bold.

| Methods | Beauty | | | Shoes | | | Phones | | | All 5 categories | | |
|---|---|---|---|---|---|---|---|---|---|---|---|---|
| | P | R | F1 | P | R | F1 | P | R | F1 | P | R | F1 |
| Bert* | 0.5355 | 0.6284 | 0.5782 | 0.5851 | 0.7020 | 0.6382 | 0.4212 | 0.5384 | 0.4726 | 0.4549 | 0.5652 | 0.5041 |
| Bert+CRF* | 0.6731 | 0.6802 | 0.6766 | 0.7302 | 0.7703 | 0.7497 | 0.5620 | 0.5923 | 0.5768 | 0.6688 | 0.6530 | 0.6608 |
| Bert+BiLSTM+CRF* | 0.7282 | 0.7481 | 0.7380 | 0.7870 | 0.8101 | 0.7984 | 0.6701 | 0.6990 | 0.6843 | 0.6892 | 0.6875 | 0.6884 |
| *No collaboration* | | | | | | | | | | | | |
| UniMIND(BART) | 0.6443 | 0.6000 | 0.6085 | 0.7711 | 0.7417 | 0.7483 | 0.7522 | 0.7406 | 0.7407 | 0.7188 | 0.6933 | 0.6978 |
| UniMIND(CPT) | 0.5994 | 0.5420 | 0.5565 | 0.7468 | 0.6836 | 0.6889 | 0.7110 | 0.6907 | 0.6959 | 0.6807 | 0.6451 | 0.6539 |
| ChatGLM | 0.7858 | 0.7797 | 0.7777 | 0.8265 | 0.8307 | 0.8248 | 0.7805 | 0.7792 | 0.7760 | 0.7968 | 0.7936 | 0.7910 |
| Chinese-Alpaca | 0.7409 | 0.7310 | 0.7316 | 0.8032 | 0.7868 | 0.7899 | 0.7363 | 0.7178 | 0.7238 | 0.7568 | 0.7378 | 0.7425 |
| *LLM assisting CRS* | | | | | | | | | | | | |
| CLLM-BCRS | 0.6502 | 0.5848 | 0.6004 | 0.7725 | 0.7171 | 0.7318 | 0.7504 | 0.7152 | 0.7246 | 0.7173 | 0.6665 | 0.6796 |
| CLLM-CCRS | 0.6101 | 0.5346 | 0.5545 | 0.7663 | 0.7218 | 0.7338 | 0.7183 | 0.6976 | 0.7018 | 0.7023 | 0.6539 | 0.6666 |
| ALLM-BCRS | 0.6255 | 0.5688 | 0.5835 | 0.7746 | 0.7120 | 0.7302 | 0.7311 | 0.7108 | 0.7159 | 0.7088 | 0.6653 | 0.6765 |
| ALLM-CCRS | 0.5730 | 0.5307 | 0.5410 | 0.7048 | 0.6658 | 0.6768 | 0.6833 | 0.6635 | 0.6691 | 0.6629 | 0.6277 | 0.6369 |
| *CRS assisting LLM* | | | | | | | | | | | | |
| BCRS-CLLM | 0.7900 | 0.7879 | 0.7824 | **0.8521** | **0.8511** | **0.8473** | **0.8222** | **0.8220** | **0.8179** | **0.8105** | **0.8065** | **0.8033** |
| CCRS-CLLM | **0.7940** | **0.7926** | **0.7878** | 0.8372 | 0.8378 | 0.8326 | 0.7911 | 0.7866 | 0.7847 | 0.7927 | 0.7897 | 0.7864 |
| BCRS-ALLM | 0.7772 | 0.7462 | 0.7542 | 0.8062 | 0.7690 | 0.7770 | 0.7662 | 0.7160 | 0.7311 | 0.7700 | 0.7310 | 0.7414 |
| CCRS-ALLM | 0.7600 | 0.7272 | 0.7354 | 0.8036 | 0.7621 | 0.7738 | 0.7392 | 0.6978 | 0.7086 | 0.7569 | 0.7168 | 0.7276 |

## 4.3 Baseline methods

For each task, baseline methods consist of typical methods, CRS methods and LLM methods.

**Typical methods.** We select typical methods for the four tasks following (Liu et al., 2023b). Specifically, we select Bert, Bert+CRF, Bert+BiLSTM+CRF as baselines for pre-sales dialogue. For user needs elicitation task, we select DiaMultiClass and DiaSeq as baselines. For user needs-based recommendation task, we choose Bert, SASRec, TG-CRS. And we select GPT-2 and KBRD as baseline methods for pre-sales dialogue generation task. For the limited space, we put the description of each typical methods in the Appendix B. We select UniMIND(BART) (Deng et al., 2023) and UniMIND(CPT) as CRS methods. For LLM methods, we select ChatGLM (Zeng et al., 2023) and Chinese-Alpaca (Cui et al., 2023). For combination of LLM and CRS, we define eight variants. We put the description of each variant in the Appendix C.

## 4.4 Evaluation metrics

We adopt the evaluation metrics used in U-NEED (Liu et al., 2023b). Specifically, we select precision, recall and f1 score as evaluation metrics for pre-sales dialogue understanding and user needs elicitation. For user needs-based recommendation task, we choose Hit@K and MRR@K. And we

adopt automatic and human evaluation for pre-sales dialogue generation task. For automatic evaluation, we use Distinct-n. And for human evaluation, we measure the informativeness and relevance of generated response. For the limited space, we put the description of each metric in the Appendix D.

## 5 Results and Analysis

We conduct extensive experiments to explore the performance of collaborations of LLMs and CRSs on four tasks. We analyze impacts of collaborations on each task in turn.

### 5.1 Impacts of collaborations on pre-sales dialogue understanding

Based on Table 1 we have the following observations: (i) *LLMs perform well in understanding user needs.* ChatGLM substantially outperforms classical baseline methods and CRSs on all metrics in all categories. The second best method is Chinese-Alpaca, which outperforms the strongest baseline method, Bert+BiLSTM+CRF, on most metrics. We attribute this to the strong capability of the LLMs for dialogue understanding. (ii) *In collaborations with CRSs, LLMs perform even better in understanding user needs.* BCRS-CLLM outperforms ChatGLM in all category and all metrics, especially in *Shoes* and *Phones*. Similarly we observe that BCRS-ALLM outperforms Chinese-Alpaca in some metrics. By carefully comparing

Table 2: Performance of baseline methods on user needs elicitation task in 3 typical categories: *Beauty*, *Fashion* and *Shoes*. Baseline results marked with * are taken from U-NEED (Liu et al., 2023b). CLLM is short for ChatGLM and ALLM is short for Chinese-Alpaca. BCRS is short for UniMIND(BART) and CCRS is short for UniMIND(CPT). The best results are highlighted in bold.

| | Beauty | | | Shoes | | | Phones | | | All 5 categories | | |
|---|---|---|---|---|---|---|---|---|---|---|---|---|
| | P | R | F1 | P | R | F1 | P | R | F1 | P | R | F1 |
| DiaMultiClass* | 0.4037 | **0.7228** | **0.5054** | 0.3361 | **0.4131** | 0.3423 | 0.4534 | **0.5212** | 0.4585 | 0.3222 | **0.4966** | 0.3662 |
| DiaSeq* | 0.4761 | 0.4272 | 0.4424 | 0.3992 | 0.3305 | 0.3498 | 0.4414 | 0.3789 | 0.3966 | 0.3555 | 0.2996 | 0.3153 |
| *No collaboration* | | | | | | | | | | | | |
| UniMIND(BART) | 0.4979 | 0.4305 | 0.4518 | 0.4367 | 0.3827 | 0.3927 | 0.5513 | 0.4973 | 0.5061 | 0.4301 | 0.3881 | 0.3943 |
| UniMIND(CPT) | 0.4022 | 0.3575 | 0.3657 | 0.4388 | 0.3774 | 0.3906 | 0.4946 | 0.4432 | 0.4515 | 0.4021 | 0.3575 | 0.3657 |
| ChatGLM | 0.4348 | 0.4057 | 0.4055 | 0.4054 | 0.3496 | 0.3621 | 0.4712 | 0.4541 | 0.4441 | 0.3683 | 0.3422 | 0.3380 |
| Chinese-Alpaca | 0.2723 | 0.2362 | 0.2475 | 0.4020 | 0.3440 | 0.3588 | 0.4946 | 0.4613 | 0.4604 | 0.3513 | 0.3132 | 0.3191 |
| *LLM assisting CRS* | | | | | | | | | | | | |
| CLLM-BCRS | **0.5106** | 0.4413 | 0.4617 | 0.4510 | 0.3837 | **0.4016** | **0.5703** | 0.4982 | **0.5168** | **0.4518** | 0.3963 | **0.4095** |
| CLLM-CCRS | 0.4128 | 0.3405 | 0.3583 | **0.4531** | 0.3735 | 0.3955 | 0.5243 | 0.4541 | 0.4737 | 0.4056 | 0.3405 | 0.3583 |
| ALLM-BCRS | 0.4702 | 0.4149 | 0.4289 | 0.4490 | 0.3827 | 0.3969 | 0.5297 | 0.4865 | 0.4863 | 0.4258 | 0.3866 | 0.3903 |
| ALLM-CCRS | 0.4043 | 0.3362 | 0.3574 | 0.4490 | 0.3661 | 0.3884 | 0.5505 | 0.4928 | 0.5009 | 0.4138 | 0.3519 | 0.3671 |
| *CRS assisting LLM* | | | | | | | | | | | | |
| BCRS-CLLM | 0.4908 | 0.4319 | 0.4450 | 0.3918 | 0.3280 | 0.3429 | 0.4923 | 0.4788 | 0.4645 | 0.4190 | 0.3723 | 0.3796 |
| CCRS-CLLM | 0.4092 | 0.3766 | 0.3783 | 0.4082 | 0.3439 | 0.3590 | 0.4495 | 0.4356 | 0.4211 | 0.4020 | 0.3710 | 0.3704 |
| BCRS-ALLM | 0.2660 | 0.2383 | 0.2428 | 0.2490 | 0.1914 | 0.2067 | 0.2568 | 0.2432 | 0.2378 | 0.2350 | 0.1904 | 0.1994 |
| CCRS-ALLM | 0.1681 | 0.1489 | 0.1539 | 0.2388 | 0.1826 | 0.1951 | 0.2027 | 0.1874 | 0.1823 | 0.2071 | 0.1632 | 0.1730 |

the results of the four combinations of "CRS assisting LLM" with the results of the four methods of "No collaboration", we find that a better performing CRS improves the performance, while a worse performing CRS degrades the performance. Since LLMs are usually very sensitive to inputs, we think that the former may provide useful reference information to complement what the LLMs do not take into account. The latter, on the other hand, may bring in noises that disturb the judgments of the LLMs. (iii) ***The improvement in understanding user needs brought by the collaboration of CRS to LLMs varies across categories.*** In the *Shoes* and *Phones* categories, BCRS-CLLM significantly outperforms ChatGLM with the collaborations of CRSs. While in the *Beauty* category, BCRS-CLLM has only a minor improvement compared to Chat-GLM. We think this may be due to the fact that user needs in the *Beauty* category are usually focused on specific attributes such as "skin type".

## 5.2 Impacts of collaborations on user needs elicitation

Based on Table 2 we have the following observations: (i) ***LLMs do not exhibit superb performance on user needs elicitation.*** On the average results of the 5 categories, UniMIND(BART) achieves the best performance, followed by the classical method DialMultiClass and UniMIND(CPT). Moreover, DiaMultiClass beats all the methods on Recall met-

rics in Beauty, Shoes, and Mobile categories. This indicates that in E-commerce pre-sales dialogue, the performance of methods that make decisions with the help of generative models, e.g. BART and LLMs, is somewhat limited. DialMultiClass doesn't require a large number of parameters and doesn't need to be trained for a long period of time. Compared to making decisions with LLMs, DialMultiClass still has considerable strengths in real-world production environments. (ii) ***Collaborations between ChatGLM and UniMIND (BART) can improve their respective performance in user needs elicitation.*** Specifically, CLLM-BCRS outperforms UniMIND(BART) on all metrics for all categories. Moreover, CLLM-BCRS beats all methods on six metrics. Similarly, BCRS–CLLM outperforms ChatGLM in all categories except *Shoes*. Specifically, BCRS-CLLM achieves a 12.3% improvement over ChatGLM on the average F1 score across all 5 categories. Based on this, we see that the output of ChatGLM are beneficial for UniMIND(BART) and vice versa. (iii) ***Chinese-Alpaca and ChatGLM exhibit major differences in their performance in collaborations with CRSs.*** Specifically, CCRS-ALLM and BCRS-ALLM achieve the worst and second–worst performance on almost all metrics in all categories. This implies that the predicted results of CRSs cause a large disruption to Chinese-Alpaca. The two combinations of CRSs assisting

Table 3: Performance of baseline methods on user needs-based recommendation task in 3 typical categories: *Beauty*, *Fashion* and *Shoes*. H@K and M@K refer to Hit@K and MRR@K. Acc. refers to accuracy. CLLM is short for ChatGLM and ALLM is short for Chinese-Alpaca. BCRS is short for UniMIND(BART) and CCRS is short for UniMIND(CPT). Since LLM methods only recommend 1 product, H@5 and M@5 cannot be calculated. The best results are highlighted in bold.

| | Beauty | | | Shoes | | | Phones | | | All 5 categories | | |
|---|---|---|---|---|---|---|---|---|---|---|---|---|
| | Acc. | H@5 | M@5 | Acc. | H@5 | M@5 | Acc. | H@5 | M@5 | Acc. | H@5 | M@5 |
| Bert | 0.0123 | 0.0185 | 0.0141 | 0.0046 | 0.0138 | 0.0072 | 0.0326 | 0.0688 | 0.0463 | 0.0118 | 0.0215 | 0.0151 |
| SASRec | 0.1108 | 0.2831 | 0.1711 | 0.0399 | 0.1121 | 0.0668 | 0.0761 | 0.2681 | 0.1475 | 0.0976 | 0.2532 | 0.1556 |
| TG-CRS | 0.1323 | 0.3354 | 0.2034 | 0.1275 | 0.2396 | 0.1692 | 0.2564 | 0.4928 | 0.3347 | 0.1744 | 0.3074 | 0.2244 |
| *No collaboration* | | | | | | | | | | | | |
| UniMIND(BART) | 0.2154 | 0.6246 | 0.3654 | 0.2458 | 0.5315 | 0.3483 | 0.3478 | 0.6449 | 0.4608 | 0.2398 | 0.5440 | 0.3510 |
| UniMIND(CPT) | 0.2554 | 0.6246 | **0.3915** | 0.2826 | 0.5438 | 0.3779 | 0.3043 | 0.6594 | 0.4460 | 0.2639 | 0.5617 | 0.3737 |
| ChatGLM | 0.2123 | - | - | 0.3810 | - | - | 0.0580 | - | - | 0.2226 | - | - |
| Chinese-Alpaca | 0.1415 | - | - | **0.3917** | - | - | 0.0036 | - | - | 0.2157 | - | - |
| *LLM assisting CRS* | | | | | | | | | | | | |
| CLLM-BCRS | 0.2462 | 0.6062 | 0.3741 | 0.2565 | 0.5376 | 0.3594 | 0.3804 | 0.7536 | 0.5204 | 0.2623 | 0.5617 | 0.3694 |
| CLLM-CCRS | **0.2585** | 0.6308 | 0.3913 | 0.2657 | 0.5438 | 0.3690 | 0.4058 | 0.7138 | 0.5319 | 0.2768 | 0.5665 | 0.3835 |
| ALLM-BCRS | 0.2554 | 0.6092 | 0.3822 | 0.2550 | 0.5223 | 0.3554 | 0.3587 | **0.7681** | 0.5196 | 0.2623 | 0.5606 | 0.3721 |
| ALLM-CCRS | 0.2430 | **0.6369** | 0.3747 | 0.2750 | **0.5515** | **0.3788** | **0.4312** | 0.7609 | **0.5959** | **0.2822** | **0.5832** | **0.3919** |
| *CRS assisting LLM* | | | | | | | | | | | | |
| BCRS-CLLM | 0.1600 | - | - | 0.2442 | - | - | 0.3478 | - | - | 0.2264 | - | - |
| CCRS-CLLM | 0.1785 | - | - | 0.2642 | - | - | 0.3043 | - | - | 0.2479 | - | - |
| BCRS-ALLM | 0.2000 | - | - | 0.2458 | - | - | 0.3297 | - | - | 0.2307 | - | - |
| CCRS-ALLM | 0.2369 | - | - | 0.2688 | - | - | 0.2754 | - | - | 0.2532 | - | - |

ChatGLM, i.e., BCRS-CLLM and CCRS-CLLM, however, perform well. We think that the differences between ChatGLM and Chinese-Alpaca in collaborating with CRSs come from their base models and fine-tuning data.

## 5.3 Impacts of collaborations on user needs-based recommendation

Based on Table 3 we have the following observations: (i) ***LLMs show the potential for user needs-based recommendation.*** Specifically, LLMs, i.e., Chinese-Alpaca and ChatGLM, achieve the best and second best performance significantly outperforming all the methods on the Accuracy in Shoes category, respectively. They also achieve performance over classical methods on the results of all 5 categories. Based on this, we think that LLMs can somewhat provide suitable recommendations when the candidate range is small (the number of candidate products in Table 3 is 20). (ii) ***With the collaboration of LLMs, the recommendation performance of CRSs can be improved.*** Specifically, on the average results across all 5 categories, ALLM-CCRS achieves 6.9%, 3.8%, and 4.9% improvements on Accuracy, Hit@5, and MRR@5, respectively, compared to UniMIND (CPT). Similarly, on average results across all 5 categories, ALLM-BCRS achieves 9.4%, 3.0%, and 6.0% improvements on Accuracy, Hit@5, and MRR@5,

respectively, when compared to UniMIND(BART). Note that LLMs provide recommendations in a different way than CRSs do. LLMs provide recommendations relying on given inputs, i.e., a recommended product is semantically related to user needs in some way. CRSs, on the other hand, model a representation of both and learn the implicit relationship between the two to compute the probability of a product being recommended. The above improvements are only that CRSs consider the representations of the recommended products given by the LLMs. We believe that collaborations between LLMs and CRSs on recommendation tasks go far beyond this and are a direction worth exploring. (iii) ***With collaborations with CRSs, LLMs can achieve comparable recommendation performance.*** Specifically, in the *Phones* category, ChatGLM and Chinese-Alpaca have very poor recommendation performance. In contrast, the four methods of "CRS assisting LLM" achieve the performance close to that of CRSs. Based on this, we think that when the recommendation performance of LLMs is very poor in a certain domain, a collaborative approach could make LLMs to achieve performance close to that of CRSs.

Table 4: Performance of baseline methods on pre-sales dialogue generation task in 3 typical categories: *Beauty*, *Fashion* and *Shoes*. CLLM is short for ChatGLM and ALLM is short for Chinese-Alpaca. BCRS is short for UniMIND(BART) and CCRS is short for UniMIND(CPT). Info. and Rel. refer to informativeness and relevance. The best results are highlighted in bold.

| | Beauty | | | Shoes | | | Phones | | | All 3 categories | | |
|---|---|---|---|---|---|---|---|---|---|---|---|---|
| | Dist-1 | Rel. | Info. | Dist-1 | Rel. | Info. | Dist-1 | Rel. | Info. | Dist-1 | Rel. | Info. |
| GPT-2 | 0.6195 | 1.8800 | 1.8400 | 0.6504 | 2.1933 | 2.1467 | 0.6305 | 1.9867 | 1.8067 | 0.6335 | 2.0200 | 1.9311 |
| KBRD | 0.5753 | 2.9933 | 2.5067 | 0.5639 | 3.3467 | 2.8800 | 0.6034 | 3.1000 | 2.7533 | 0.5809 | 3.1467 | 2.7133 |
| *No collaboration* | | | | | | | | | | | | |
| UniMIND(BART) | 0.8611 | 3.7933 | 3.3600 | 0.8897 | 3.9133 | 3.6933 | 0.8299 | 3.6667 | 3.6200 | 0.8521 | 3.7911 | 3.5578 |
| UniMIND(CPT) | 0.8578 | 3.7333 | 3.2600 | 0.9080 | 3.9333 | 3.7800 | 0.8472 | **3.8000** | **3.7400** | 0.8583 | 3.8222 | 3.5933 |
| ChatGLM | 0.8169 | 3.7467 | 3.4533 | 0.8594 | 3.7000 | 3.4400 | 0.8411 | 3.6200 | 3.4533 | 0.8152 | 3.6889 | 3.4489 |
| Chinese-Alpaca | **0.9131** | 3.6667 | 3.2467 | **0.9124** | 3.8733 | 3.6667 | **0.9109** | 3.6733 | 3.6133 | **0.8927** | 3.7378 | 3.5089 |
| *LLM assisting CRS* | | | | | | | | | | | | |
| CLLM-BCRS | 0.8686 | 3.8467 | 3.4933 | 0.8986 | 3.9133 | 3.7600 | 0.8460 | 3.7267 | 3.6000 | 0.8611 | 3.8289 | **3.6178** |
| CLLM-CCRS | 0.8633 | 3.7667 | **3.5000** | 0.8975 | 3.9267 | 3.7200 | 0.8568 | 3.6267 | 3.5267 | 0.8587 | 3.7734 | 3.5822 |
| ALLM-BCRS | 0.8700 | 3.7667 | 3.4000 | 0.8959 | 3.9200 | **3.7867** | 0.8505 | 3.7333 | 3.6533 | 0.8650 | 3.8067 | 3.6133 |
| ALLM-CCRS | 0.8792 | 3.8467 | 3.4467 | 0.9086 | **3.9933** | **3.7867** | 0.8640 | 3.6400 | 3.5933 | 0.8699 | 3.8267 | 3.6089 |
| *CRS assisting LLM* | | | | | | | | | | | | |
| BCRS-CLLM | 0.8371 | 3.6867 | 3.3867 | 0.8735 | 3.9400 | 3.7667 | 0.8544 | 3.6733 | 3.5800 | 0.8365 | 3.7667 | 3.5778 |
| CCRS-CLLM | 0.8324 | 3.5000 | 3.0867 | 0.8714 | 3.9200 | 3.7800 | 0.8355 | 3.5933 | 3.4533 | 0.8256 | 3.6711 | 3.4400 |
| BCRS-ALLM | 0.8910 | 3.6933 | 3.2200 | 0.9011 | 3.7867 | 3.6133 | 0.9040 | 3.6067 | 3.5400 | 0.8804 | 3.6956 | 3.4578 |
| CCRS-ALLM | 0.8898 | **3.8600** | 3.3400 | 0.8951 | 3.9333 | 3.6733 | 0.8988 | 3.7400 | 3.5667 | 0.8826 | **3.8444** | 3.5267 |

## 5.4 Impacts of collaborations on pre-sales dialogue generation

Based on Table 4 we have the following observations: (i) ***CRSs and LLMs show comparable performance in pre-sales dialogue generation.*** In Shoes, Beauty, and Phones categories, Chinese-Alpaca achieves the best performance on Dist-1. This indicates that Chinese-Alpaca can generate more diverse responses. While in most cases, the responses generated by CRSs are more relevant and informative than those generated by LLMs. In addition, neither the LLMs nor the CRSs generate responses that beat the ground truth responses provided by customer service staff. (ii) ***Collaborations between LLMs and CRSs show marginal effects on pre-sales dialogue generation.*** Specifically, the methods of collaborations between LLMs and CRSs, i.e., "LLM assisting CRS" and "CRS assisting LLM", achieve the best performance on most of the metrics. However, the improvement from collaboration is marginal compared to CRSs or LLMs. We believe this may be due to the fact that LLMs and UniMIND are relatively close in their approaches to generating responses, i.e., both are based on pre-trained language models and prompts. Therefore, collaborations between two similar approaches does not have much impact. In future work, we plan to consider CRSs that focus on generating persuasive reasons for recommendations, e.g., NTRDs that introduce words related to the

recommended items in the decoding process. Intuitively, collaborations between such CRSs and LLMs may work out well.

## 6 Conclusions and Future Work

In this paper, we investigated the integration of conversational recommender systems (CRS) and large language models (LLM) in E-commerce pre-sales dialogues. Specifically, we proposed two collaboration strategies: "CRS assisting LLM" and "LLM assisting CRS". We evaluate the effectiveness of these collaborations between two LLMs and two CRSs on four tasks related to E-commerce pre-sales dialogues. Through extensive experiments and careful analysis, we found that the collaboration between CRS and LLM can be highly effective in certain scenarios, providing valuable insights for both research and practical applications in E-commerce pre-sales dialogues. Additionally, our findings can inspire exploration of collaborations between general large models and private small models in other domains and areas.

For future work, we plan to examine the collaboration between LLMs and CRSs across various categories. For instance, a CRS within the *Shoes* category could provide information to a LLM in the *Fashion* category, resulting in a recommended combination such as a dress, pants, and shoes.

## Limitations

One limitation of our work is that our findings may only apply to large language models (LLMs) around 7B parameters. In this work, we select two LLMs that are widely used, namely ChatGLM-6B and Chinese-Alpaca-7B. Related work reveals that there is some variation in the capability of LLMs with different parameter sizes (Wei et al., 2022). For LLMs with more parameters, such as 100B, more GPU resources and time are needed to explore the effects of combining CRSs and LLMs. In addition, LLMs are constantly being updated. Recently ChatGLM2-6B and Chinese-Alpaca2-13B have been open sourced.[3] They show better performance than ChatGLM-6B and Chinese-Alpaca-7B on multiple benchmarks, and may have higher results on E-commerce pre-sales dialogues as well. However, we believe that the combination of LLMs and CRSs is still worth researching.

## Ethics Statement

In this paper, we explore the effectiveness of combining LLM and CRS on e-commerce pre-sales conversations. We are committed to using and modify U-NEED dataset only for research purposes and not for commercial exploitation. Our proposed approach does not generate harmful content or raise ethical issues.

## Acknowledgments

We thank the anonymous reviewers for their helpful comments. This work is supported by the National Key Research and Development Program (No. 2022YFF0902100) and National Natural Science Foundation of China (No. 62076081 and No. 61936010).

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

## A Pre-sales Dialogue Tasks

To evaluate the performance of CRSs, LLMs and their collaborations in E-commerce scenarios, we adopt four challenging tasks proposed in U-NEED dataset (Liu et al., 2023b). The four challenging tasks are: (i) pre-sales dialogue understanding (ii) user needs elicitation (iii) user needs-based recommendation and (iv) pre-sales dialogue generation.

Pre-sales dialogue understanding aims to understand utterances of both users and customer service staff. First, identify the attributes that are related to products. And second, extract preferences related to the identified attributes. For example, when a user says, "Which of your thermal underwear is the warmest? Recommend one?" This task aims to obtain semantic frames { ("Functional requirement", "Warmest"), ("Category", "Thermal underwear") }, where "Functional requirement" and "Category" are attributes related to products. "Warmest" and "Thermal underwear" are preferences.

User needs elicitation aims to select attributes that can elicit more information about user needs.

The inputs for this task are the dialogue context and the identified user needs, i.e., { ("Functional requirement", "Warmest"), ("Category", "Thermal underwear") }. The output is a set of attributes, e.g., {"Category", "Price"}.

User needs-based recommendation aims to recommend products that satisfy explicit and implicit user needs. Explicit user needs refer to the needs and preferences expressed by the user in the ongoing dialogue, i.e., { ("Functional requirement", "Warmest"), ("Category", "Thermal underwear") }. Implicit user needs are related to user behaviors outside of the pre-sales dialogue. Users usually view some items before starting a dialogue with the customer service staff. In addition, they browse through items while talking to customer service staff. Such behaviors can reflect implicit user needs to some extent. The inputs to this task are explicit and implicit user needs, i.e., identified semantic frames and user behaviors, and the output is a collection of items.

Pre-sales dialogue generation aims to generate a response based on given information about user needs. Information consists of a collection of attributes and a collection of items. The collection of attributes is the output of the user needs elicitation task, i.e., the attributes that may elicit more information about the user's needs. The collection of items is the output of the user needs-based recommendation task, i.e., items that satisfy the current user needs. The inputs to this task are the dialogue context, the collection of attributes and the collection of items. The output is a response, which may be a query asking a question about an attribute, e.g.,"What are your requirements for the thermal underwear?" or a recommendation reason for recommending an item, e.g., "Recommended item: 655398643290. This one is seamless and made of double-sided fleece. You can take a look."

## B Baselines for Pre-sales Dialogue Tasks

Following Liu et al. (2023b), we (i) select Bert (Devlin et al., 2019), Bert+CRF (Souza et al., 2019) and Bert+BiLSTM+CRF (Dai et al., 2019) as baselines for the pre-sales dialogue understanding task; (ii) select DiaMultiClass (Li et al., 2020) and DiaSeq (Li et al., 2020) as baselines for the user needs elicitation task; (iii) select Bert (Devlin et al., 2019), SASRec (Kang and McAuley, 2018) and TG-CRS (Zhou et al., 2020) as baselines for the user needs-based recommendation task; (iv) and se-

lect GPT-2 (Radford et al., 2019) and KBRD (Chen et al., 2019) as baselines for the pre-sales dialogue generation task.

For the pre-sales dialogue understanding task, Bert, Bert+CRF, and Bert+BiLSTM+CRF adopt the sequence labeling approach to identify the semantic frames. Bert considers only the representation of the input utterance. Bert+CRF takes into account the sequential relationships of the predicted tags in addition to the representation of the input utterance. Whereas Bert+BiLSTM+CRF adds bidirectional information encoding after obtaining the representation of the input and considers the sequential relationships of the predicted tags to compute the probability of each tag.

For the user needs elicitation task, DiaMultiClass and DiaSeq employ a multi-label classification approach to determine the collection of attributes.DiaMultiClass computes the probability of each attribute based on the representation of the inputs.DiaSeq computes the probability of each attribute based on the sequential relationship between semantic frames.

The baseline for the user needs-based recommendation task is Bert, SASRec and TG-CRS. Bert calculates the probability of an item based on the representation of the input. SASRec calculates the probability of an item based on the sequential relationships of user behaviors. TG-CRS considers both dialogue context and sequential user behaviors.

For the pre-sales dialogue generation task, the baselines are GPT-2 and KBRD. GPT-2 is a commonly used pre-trained language model for the dialogue generation task. KBRD utilizes a switching mechanism to introduce tokens related to the recommended items during the decoding of responses.

## C Combinations of LLMs and CRSs

We define four variants of LLM assists CRS:

- CLLM-BCRS refers that ChatGLM assists BART-based CRS.

- CLLM-CCRS refers that ChatGLM assists CPT-based CRS.

- ALLM-BCRS refers that Chinese-Alpaca assists BART-based CRS.

- ALLM-CCRS refers that Chinese-Alpaca assists BART-based CRS.

We define four variants of CRS assists LLM:

- BCRS-CLLM refers that BART-based CRS assists ChatGLM.

- CCRS-CLLM refers that CPT-based CRS assists ChatGLM.

- BCRS-ALLM refers that BART-based CRS assists Chinese-Alpaca.

- CCRS-ALLM refers that CPT-based CRS assists Chinese-Alpaca.

## D   Evaluation Metrics

Following Liu et al. (2023b), for the pre-sales dialogue understanding and user needs elicitation tasks, we set Precision, Recall and F1 score as evaluation metrics. Precision is the proportion of correctly selected tags to the total number of selected tags. Recall is the ratio of correctly selected tags to the original number of correct tags. The F1 score is calculated by taking the harmonic mean of precision and recall.

Regarding the user needs-based recommendation task, the evaluation metrics in U-NEED (Liu et al., 2023b) are Hit@10, Hit@50 and MRR@50. Due to the limitation of the input length of LLMs, where each product contains attributes and attribute values, we can provide a maximum of 20 candidate products. Therefore, in order to compare whether the collaborative approach improves the performance of CRSs, we measure Accuracy (Hit@1), Hit@5 and MRR@5. The Hit@K metric represents the proportion of relevant items that are present in the top-K results out of all the relevant items. The MRR@K score is determined by taking the average of the reciprocal ranks of the top-K items in a ranking. If an item does not appear in the top-K positions, its reciprocal rank is set to 0.

The evaluation metrics for the pre-sales dialogue generation task are Distinct@1, Informativeness and Relevance. Distinct@1 is computed as the average of the fraction of distinct 1-grams out of all 1-grams in a response. Distinct@1 measures the diversity of generated responses. Informativeness and relevance are for human evaluation. We randomly sample 100 dialogues and we recruit 12 annotators to evaluate 1400 responses from 14 methods on these 100 dialogues. Informativeness is calculated as the average informativeness of all generated responses. Relevance is determined as the average relevance degree of all generated responses.

The annotators evaluate the extent to which a generated response includes information about the product, as compared to the ground truth. The score of informativeness and relevance ranges from 1 to 5, and we calculate the average score from all annotators to obtain the final score.

## E   Implementation Details

We implement CRSs based on UniMIND.[4] The code is available online.[5] For CRSs, we use a NVIDIA A100-SXM4-80GB gpu and train model for 10 epochs, with a duration of approximately 12 hours. For LLMs, we use a NVIDIA A100-SXM4-80GB gpu and train model for 3 epochs, with a duration of approximately 9 hours.

## F   Examples of Fine-tuning LLMs

We give examples of the instructions, inputs, and outputs used to fine-tune the LLMs for each task in Tables 5, 6, 7, and 8, respectively.

## G   Examples of Collaborations of CRSs and LLMs

We show examples of inputs (instructions) for collaborations between CRSs and LLMs on pre-sales dialogue understanding and generation tasks in Fig. 4 and Fig. 5, respectively.

---

[4] https://github.com/dengyang17/UniMIND
[5] https://github.com/LeeeeoLiu/LLM-CRS

**No collaboration**

**Instruction and input for LLMs**

Instruction:根据服装行业售前对话，识别当前用户或者客服输入中涉及的商品相关的属性值和对应的属性。\n针对用户输入，需要识别出属性和属性值。客服输入中属性值可能为空。(Based on pre-sales dialogue in the fashion domain, identify the item related attribute values and corresponding attributes mentioned in the current input of user or customer service rep. \nUser input requires identification of attributes and their values, while customer service rep input may have empty attribute values.)

Input:售前对话：[Round 0]\n用户：你们家保暖性强一点的秋衣是哪种？推荐一下\n客服：亲亲，你是要秋衣还是要保暖衣呢\n当前输入：用户：保暖，冬天穿的，怕冷 (Pre-sales dialogue: [Round 0]\nUser: Which of your thermal underwear is the warmest? Recommend one? \n Customer Service Rep: What are your requirements for the thermal underwear?\n[Round 1] \nCurrent User: I 'd like thermal underwear for winter wear, since I feel the cold. )

Output:功能需求：保暖；使用时间：冬天 (Functional requirement : thermal ; Usage Period : winter )

**Prompt for CRSs**

[user]你们家保暖性强一点的秋衣是哪种？推荐一下[understand]功能需求：保暖性强；品类：秋衣。 [system]亲亲，你是要秋衣还是要保暖衣呢[understand]品类：秋衣；品类：保暖衣[user]保暖，冬天穿的，怕冷。识别用户需求信息：

([User]Which of your thermal underwear is the warmest? Recommend one? [Understand]Functional requirement warmest; Category: thermal underwear. [System]What are your requirements for the thermal underwear? [Understand]Category: thermal underwear [User] I 'd like thermal underwear for winter wear, since I feel the cold. Identify the user requirements:)

**Collaborations of CRS and LLM**

*CRS assisting LLM*

Instruction:根据服装行业售前对话，*并参考对话式推荐系统的识别结果*，识别当前用户或者客服输入中涉及的商品相关的属性。\n针对用户输入，需要识别出属性和属性值。客服输入中属性值可能为空。(Based on pre-sales dialogue in the fashion domain and *the identification results of CRS*, identify the item related attribute values and corresponding attributes mentioned in the current input of user or customer service rep. \nUser input requires identification of attributes and their values, while customer service rep input may have empty attribute values.)

Input:售前对话：[Round 0]\n用户：你们家保暖性强一点的秋衣是哪种？推荐一下\n客服：亲亲，你是要秋衣还是要保暖衣呢\n当前输入：用户：保暖，冬天穿的，怕冷*对话式推荐系统识别的结果：肌肤问题：刺痛；肤质：发痒*(Pre-sales dialogue: [Round 0]\nUser: Which of your thermal underwear is the warmest? Recommend one? \n Customer Service Rep: What are your requirements for the thermal underwear?\n[Round 1] \nCurrent Input: User: I 'd like thermal underwear for winter wear, since I feel the cold. \n*The identification results of CRS: skin issue: stinging; skin type: itching*)

Output:功能需求：保暖；使用时间：冬天 (Functional requirement : thermal ; Usage Period : winter )

*LLM assisting CRS*

[user]你们家保暖性强一点的秋衣是哪种？推荐一下[understand]功能需求：保暖性强；品类：秋衣。 [system]亲亲，你是要秋衣还是要保暖衣呢[understand]品类：秋衣；品类：保暖衣[user]保暖，冬天穿的，怕冷*[LLM]功能需求：保暖；季节：冬天。*识别用户需求信息：

([User]Which of your thermal underwear is the warmest? Recommend one? [Understand]Functional requirement warmest; Category: thermal underwear. [System]What are your requirements for the thermal underwear? [Understand]Category: thermal underwear [User] I 'd like thermal underwear for winter wear, since I feel the cold. *[LLM] Functional requirement: thermal ; Season: winter.* Identify the user requirements:)

Figure 4: An example of collaboration between CRS and LLM on the pre-sales dialogue understanding task. Left side displays data used to fine-tune a LLM and train a CRS independently. The right side shows two cases of combining the two. Collaboration content is highlighted in red italics.

**No collaboration**

**Instruction and input for LLMs**

Instruction:根据服装行业售前对话中已获取的信息、引导用户需求的属性、满足用户需求的商品信息，生成回应用户需求且用户容易理解的通俗回复。(Based on the information obtained, user guiding attributes and items that meet user requirements in pre-sale dialogue in the fashion domain, generate user-friendly response that address user requirements.)

Input:售前对话：[Round 0]\n用户：你们家保暖性强一点的秋衣是哪种？推荐一下\n客服：亲亲，你是要秋衣还是要保暖衣呢\n[Round 1]\n用户：保暖，冬天穿的，怕冷\n已获取的用户需求偏好信息：功能需求：保暖性强、保暖；品类：秋衣；季节：冬天\n满足用户需求的商品信息：商品A满足用户需求，A的价格区间是低，功能需求是保暖，季节是冬、秋，材质是棉质，款式是宽松，类目是保暖套装(Pre-sales dialogue: [Round 0]\nUser: Which of your thermal underwear is the warmest? Recommend one? Customer Service Rep: What are your requirements for the thermal underwear? \n[Round 1]\nUser: I 'd like thermal underwear for winter wear, since I feel the cold. \nObtained user preference: Functional requirement : warmest, thermal; Category: thermal underwear; Season: winter \n items that meet user requirements: item A meet user requirements, the price range is low, the functional requirement is thermal, the seasons are winter and autumn, the material is cotton, the style is loose, the category is thermal set)

Output: 这款无痕双面绒的，亲看一下哈(This one is seamless and made of double-sided fleece. You can take a look.)

**Prompt for CRSs**

[user]你们家保暖性强一点的秋衣是哪种？推荐一下[understand]功能需求：保暖性强；品类：秋衣[system]亲亲，你是要秋衣还是要保暖衣呢[understand]品类：秋衣；品类：保暖衣[user]保暖，冬天穿的，怕冷。[understand]需求：功能需求：保暖；季节：冬天。[recommend]655398643290。生成系统回复：([User]Which of your thermal underwear is the warmest? Recommend one?[Understand]Functional requirement warmest; Category: thermal underwear [System]What are your requirements for the thermal underwear? [Understand]Category: thermal underwear [User] I 'd like thermal underwear for winter wear, since I feel the cold. [Understand]Functional requirement : thermal; Usage Period : winter [recommend]655398643290. System response generation:)

**Collaborations of CRS and LLM**

*CRS assisting LLM*

Instruction:根据服装行业售前对话中已获取的信息、引导用户需求的属性、满足用户需求的商品信息，*并参考对话式推荐系统生成的回复*，生成回应用户需求且用户容易理解的通俗回复。(Based on the information obtained, user guiding attributes and items that meet user requirements in pre-sale dialogue in the fashion domain, *as well as the response generated by CRS*, generate user-friendly response that address user requirements.)

Input:售前对话：[Round 0]\n用户：你们家保暖性强一点的秋衣是哪种？推荐一下\n客服：亲亲，你是要秋衣还是要保暖衣呢\n[Round 1]\n用户：保暖，冬天穿的，怕冷\n已获取的用户需求偏好信息：功能需求：保暖性强、保暖；季节：冬天\n满足用户需求的商品信息：商品A满足用户需求，A的价格区间是低，功能需求是保暖，季节是冬、秋，材质是棉质，款式是宽松，类目是保暖套装\n*对话式推荐系统生成的回复：亲亲需要什么功效的呢*Pre-sales dialogue: [Round 0]\nUser: Which of your thermal underwear is the warmest? Recommend one? Customer Service Rep: What are your requirements for the thermal underwear? \n[Round 1]\nUser: I 'd like thermal underwear for winter wear, since I feel the cold. \nObtained user preference: Functional requirement : warmest, thermal; Category: thermal underwear; Season: winter \n items that meet user requirements: item A meet user requirements, the price range is low, the functional requirement is thermal, the seasons are winter and autumn, the material is cotton, the style is loose, the category is thermal set\n *Response generated by CRS: What functions do you need for the product?*)

Output: 这款无痕双面绒的，亲看一下哈(This one is seamless and made of double-sided fleece. You can take a look.)

*LLM assisting CRS*

[user]你们家保暖性强一点的秋衣是哪种？推荐一下[understand]功能需求：保暖性强；品类：秋衣[system]亲亲，你是要秋衣还是要保暖衣呢[understand]品类：秋衣；品类：保暖衣[user]保暖，冬天穿的，怕冷。[understand]需求：功能需求：保暖；季节：冬天。[recommend]655398643290*[LLM]您看下这款哦，这款是有4D防菌抗菌防螨的效果的。*生成系统回复：([User]Which of your thermal underwear is the warmest? Recommend one?[Understand]Functional requirement warmest; Category: thermal underwear [System]What are your requirements for the thermal underwear? [Understand]Category: thermal underwear [User] I 'd like thermal underwear for winter wear, since I feel the cold. [Understand] Functional requirement : thermal; Usage Period : winter [recommend]655398643290*[LLM] Take a look at it. It has a 4D anti-bacterial and anti-mite function.* System response generation:)

Figure 5: An example of collaboration between CRS and LLM on the pre-sales dialogue generation task. Left side displays data used to fine-tune a LLM and train a CRS independently. The right side shows two cases of combining the two. Collaboration content is highlighted in red italics.

Table 5: Three examples of fine-tuning LLMs for pre-sales dialogue understanding task in Appliance, Beauty and Fashion categories.

| | Example | Translation |
|---|---|---|
| Instruction | 结合大家电行业售前对话，识别当前用户或者客服输入中涉及的商品相关的属性值和对应的属性。针对用户输入，需要识别出属性和属性值。客服输入中属性值可能为空。 | Combined with the pre-sales dialogue in Appliance category, identify the product-related attribute values and corresponding attributes involved in the input of the current user or customer service. For user input, attributes and attribute values need to be identified. The attribute value in the customer service input may be empty. |
| Input | 售前对话：用户：帮我推荐一款普通洗衣机，性价比高，皮实耐用的，不要烘干功能的 当前输入：客服：波轮还是滚筒呢 | Pre-sale dialogue: User: Help me recommend an ordinary washing machine with high cost performance, durable leather, and no drying function Current input: Customer service: Wave wheel or drum |
| Output | 洗衣机类型：波轮;洗衣机类型：滚筒 | washing machine type: wave wheel; washing machine type: drum |
| Instruction | 结合美妆行业售前对话，识别当前用户或者客服输入中涉及的商品相关的属性值和对应的属性。针对用户输入，需要识别出属性和属性值。客服输入中属性值可能为空。 | Combined with the pre-sales dialogue in Beauty category, identify the product-related attribute values and corresponding attributes involved in the input of the current user or customer service. For user input, attributes and attribute values need to be identified. The attribute value in the customer service input may be empty. |
| Input | 售前对话：用户：你们店有没有套装[SEP]客服：您想要什么类型的呢[SEP]用户：补水[SEP]客服：水乳吗亲亲[SEP]用户：嗯，需要水乳[SEP]客服：亲亲需要祛痘的吗[SEP]用户：需要祛痘的 当前输入：客服：亲亲是想解决红肿痘痘还是闭口的么 | Pre-sale dialogue: User: Do you have a set in your store? [SEP] Customer service: What type do you want? [SEP] User: Hydration [SEP] Customer service: Do you want milk? Kiss [SEP] User: Well, I need water Milk[SEP]Customer service: Kiss, do you need to get rid of acne[SEP]User: Need to get rid of acne Current input: Customer service: Do you want to solve red, swollen, pimples or keep your mouth shut? |
| Output | 肌肤问题：红肿痘痘;肌肤问题：闭口 | skin problem: redness, swelling and acne; skin problem: shut up |
| Instruction | 结合鞋类行业售前对话，识别当前用户或者客服输入中涉及的商品相关的属性值和对应的属性。针对用户输入，需要识别出属性和属性值。客服输入中属性值可能为空。 | Combined with the pre-sales dialogue in Fashion category, identify the product-related attribute values and corresponding attributes involved in the input of the current user or customer service. For user input, attributes and attribute values need to be identified. The attribute value in the customer service input may be empty. |
| Input | 售前对话：用户：还有没有其他款推荐[SEP]客服：要什么材质的呢亲爱哒[SEP]用户：就是冰丝的那种[SEP]客服：仅发送商品链接[SEP]客服：这个是莫代尔材质冰丝触感的[SEP]客服：仅发送商品链接[SEP]客服：这个是冰丝材质的[SEP]用户：冬天了穿哪款好点 当前输入：用户：我儿子就喜欢穿金利来这个牌子的 | Pre-sale dialogue: User: Do you have any other recommendations? [SEP] Customer service: What material do you want, dear? [SEP] User: It's the ice silk one [SEP] Customer service: Only send product links [SEP] Customer service : This is made of modal material with ice silk touch [SEP] Customer service: Only send product links [SEP] Customer service: This is made of ice silk [SEP] User: Which one is better to wear in winter Current input: User: My son I like to wear the brand of Jinlilai. |
| Output | 人群：儿子;品牌：金利来 | Crowd: Son; Brand: Goldlion |

Table 6: Three examples of fine-tuning LLMs for user needs elicitation task in Appliance, Beauty and Fashion categories.

|  | Example | Translation |
| --- | --- | --- |
| Instruction | 依据大家电行业售前对话，选择一系列的属性，来引导用户提供更多关于需求的偏好信息。结果中可以包含属性值，也可以不包含属性值。 | According to the pre-sales dialogue in Appliance category, select a series of attributes to guide users to provide more preference information about needs. Attribute values may or may not be included in the result. |
| Input | 售前对话：用户：帮我推荐一款普通洗衣机，性价比高，皮实耐用的，不要烘干功能的[SEP]客服：波轮还是滚筒呢[SEP]用户：滚筒的[SEP]用户：功能简单的[SEP]客服：仅发送商品链接[SEP]客服：仅发送商品链接[SEP]用户：有小天鹅的吗[SEP]客服：(1)专属净柔洗程序，柔和洗护爱衣，独特的全方位按摩，如同手洗般轻柔、揉搓间为衣物重塑洁净与柔软；(2)95度高温煮洗，扫净藏于衣物纤维中的病毒细菌，长效杀菌灭毒，99.9%健康除菌(3)wifi手机远程控制，随时随地，想穿就穿(4)特色羽绒服洗，分多段进水，洗涤节拍柔和，预防羽绒服漂浮水面或破损，洗护均匀，贴心呵护(5)BLDC变频电机，脱水更快更彻底，洁净少残留[SEP]用户：波轮的哪款性价比高？皮实耐用 | Pre-sale conversation: User: Help me recommend an ordinary washing machine with high cost performance, durable leather, and no drying function [SEP] Customer service: Wave wheel or drum [SEP] User: Drum [SEP] User: Simple function [SEP] Customer service: Only send product links [SEP] Customer service: Only send product links [SEP] User: Do you have Little Swan? Azimuth massage, as gentle as washing by hand, reshape the cleanliness and softness of the clothes between rubbing; (2)Boil and wash at 95 degrees high temperature, sweep away the viruses and bacteria hidden in the fibers of the clothes, long-term sterilization and disinfection, 99.9% healthy sterilization (3)Wifi mobile phone remote control , anytime, anywhere, you can wear it as you want (4)Wash the special down jacket, enter the water in multiple stages, the washing cycle is soft, prevent the down jacket from floating on the water or damage, even washing and care, caring (5)BLDC inverter motor, dehydration is faster and more thorough, clean and less residue [SEP ] User: Which one of the wave wheel is more cost-effective? Durable |
| Output | 价位 | Price |
| Instruction | 依据美妆行业售前对话，选择一系列的属性，来引导用户提供更多关于需求的偏好信息。结果中可以包含属性值，也可以不包含属性值。 | According to the pre-sales dialogue in Beauty category, select a series of attributes to guide users to provide more preference information about needs. Attribute values may or may not be included in the result. |
| Input | 售前对话：用户：你们店有没有套装 | Pre-sale dialogue: User: Do you have any suits in your store? |
| Output | 功效 | Efficacy |
| Instruction | 依据服装行业售前对话，选择一系列的属性，来引导用户提供更多关于需求的偏好信息。结果中可以包含属性值，也可以不包含属性值。 | According to the pre-sales dialogue in Fashion category, select a series of attributes to guide users to provide more preference information about needs. Attribute values may or may not be included in the result. |
| Input | 售前对话：用户：还有没有其他款推荐 | Pre-sale dialogue: User: Do you have any other recommendations? |
| Output | 材质 | Material |

Table 7: An example of fine-tuning LLMs for user needs-based recommendation task in Fashion category.

| | Example | Translation |
|---|---|---|
| Instruction | 根据服装行业售前对话中用户表达的需求和偏好信息以及候选商品信息，从候选商品A-T中选择最有可能满足用户需求、偏好的商品推荐给用户。 | According to the demand and preference information expressed by the user in the pre-sales dialogue in Fashion category and the candidate product information, the product that is most likely to meet the user's needs and preferences is selected from the candidate products A-T and recommended to the user. |
| Input | 售前对话：用户：还有没有其他款推荐[SEP]客服：要什么材质的呢亲爱哒[SEP]用户：就是冰丝的那种 各候选商品对应的属性和属性值：A的价格区间是高，功能需求是舒适，季节是夏，性别是男，服装厚度是薄款，材质是冰丝、棉质、莫代尔，款式是平角、无痕、简单，类目是男平角内裤[SEP]B的价格区间是高[SEP]C的价格区间是高[SEP]D的价格区间是高，性别是男，类目是睡衣/家居服套装[SEP]E的价格区间是高[SEP]F的价格区间是中，功能需求是保暖，季节是秋，性别是女，类目是保暖套装[SEP]G的价格区间是高[SEP]H的价格区间是高[SEP]I的价格区间是高，功能需求是保暖，性别是女，服装厚度是薄款，类目是保暖套装[SEP]J的价格区间是高[SEP]K的价格区间是高[SEP]L的价格区间是高[SEP]M的价格区间是高[SEP]N的价格区间是高[SEP]O的价格区间是高[SEP]P的价格区间是高[SEP]Q的价格区间是高[SEP]R的价格区间是高[SEP]S的价格区间是高[SEP]T的价格区间是高，款式是v领，类目是保暖套装 | Pre-sale dialogue: User: Do you have any other recommendations [SEP] Customer service: What material do you want? Dear [SEP] User: It is the kind of ice silk The attributes and attribute values corresponding to each candidate product: A The price range is high, the functional requirement is comfortable, the season is summer, the gender is male, the clothing thickness is thin, the material is ice silk, cotton, modal, the style is boxer, no trace, simple, and the category is men's boxer underwear [SEP] The price range of B is high [SEP] The price range of C is high [SEP] The price range of D is high, the gender is male, and the category is pajamas/home service sets [SEP] The price range of E is high [ The price range of SEP]F is medium, the functional requirement is to keep warm, the season is autumn, the gender is female, and the category is thermal suits. The price range of [SEP]G is high. The price range of [SEP]H is high.[SEP]I The price range is high, the functional requirement is to keep warm, the gender is female, the clothing thickness is thin, and the category is thermal suit [SEP]J, the price range is high[SEP]K, the price range is high[SEP]L It is high [SEP] the price range of M is high [SEP] the price range of N is high [SEP] the price range of O is high [SEP] the price range of P is high [SEP] the price range of Q is high [SEP] The price range of R is high [SEP] the price range of S is high [SEP] the price range of T is high, the style is v-neck, and the category is thermal suit |
| Output | A | A |

Table 8: Three examples of fine-tuning LLMs for pre-sales dialogue generation task in Appliance, Beauty and Fashion categories.

| | Example | Translation |
|---|---|---|
| Instruction | 根据大家电行业售前对话中已获取的信息、引导用户需求的属性、满足用户需求的商品信息，生成回应用户需求且用户容易理解的通俗回复。 | According to the information obtained in the pre-sales dialogue in Appliance category, the attributes that guide the user's needs, and the product information that meets the user's needs, generate a popular reply that responds to the user's needs and is easy for the user to understand. |
| Input | 售前对话：用户：帮我推荐一款普通洗衣机，性价比高，皮实耐用的，不要烘干功能的[SEP]客服：波轮还是滚筒呢[SEP]用户：滚筒的[SEP]用户：功能简单的[SEP]客服：仅发送商品链接[SEP]客服：仅发送商品链接[SEP]用户：有小天鹅吗[SEP]客服：(1)专属净柔洗程序，柔和洗护爱衣，独特的全方位按摩，如同手洗般轻柔、揉搓间为衣物重塑洁净与柔软；(2)95度高温煮洗，扫净藏于衣物纤维中的病毒细菌，长效杀菌灭毒，99.9%健康除菌(3)wifi手机远程控制，随时随地，想穿就穿(4)特色羽绒服洗，分多段进水，洗涤节拍柔和，预防羽绒服漂浮水面或破损，洗护均匀，贴心呵护(5)BLDC变频电机，脱水更快更彻底，洁净少残留[SEP]用户：波轮的哪款性价比高？皮实耐用 已获取的用户需求偏好信息：品类：洗衣机、功能需求：皮实耐用、不要烘干、功能简单、皮实耐用、款式：性价比高、性价比高、洗衣机类型：滚筒、波轮、品牌：小天鹅 引导用户需求的属性：价位： | Pre-sale conversation: User: Help me recommend an ordinary washing machine with high cost performance, durable leather, and no drying function [SEP] Customer service: Wave wheel or drum [SEP] User: Drum [SEP] User: [SEP] Customer service with simple functions: only send product links [SEP] customer service: only send product links [SEP] user: do you have Little Swan? The all-round massage is as gentle as hand washing, and the rubbing will reshape the cleanliness and softness of the clothes; (2)Boil and wash at 95 degrees high temperature, sweep away the virus bacteria hidden in the fibers of the clothes, long-term sterilization and disinfection, 99.9% healthy sterilization (3)wifi mobile phone Remote control, you can wear it anytime, anywhere (4)Special down jacket washing, multi-stage water intake, gentle washing cycle, prevent down jacket from floating or damaged, even washing and care, caring (5)BLDC frequency conversion motor, dehydration is faster and more thorough, clean and less residue [SEP] User: Which one of the wave wheel is more cost-effective? Durable Leather Acquired user demand preference information: category: washing machine, functional requirements: durable leather, no drying, simple function, durable leather, style: cost-effective, cost-effective, washing machine type: drum, pulsator, brand: Little Swan Attributes to guide user needs: Price: |
| Output | 预算多少呢亲 | How much is the budget? |
| Instruction | 根据美妆行业售前对话中已获取的信息、引导用户需求的属性、满足用户需求的商品信息，生成回应用户需求且用户容易理解的通俗回复。 | According to the information obtained in the pre-sales dialogue in Beauty category, the attributes that guide the user's needs, and the product information that meets the user's needs, generate a popular reply that responds to the user's needs and is easy for the user to understand. |
| Input | 售前对话：用户：你们店有没有套装 已获取的用户需求偏好信息：品类：套装 引导用户需求的属性：功效： | Pre-sales dialogue: User: Do you have suits in your store Acquired user demand preference information: Category: Set Attributes to guide user needs: Function: |
| Output | 您想要什么类型的呢 | what type do you want |
| Instruction | 根据服装行业售前对话中已获取的信息、引导用户需求的属性、满足用户需求的商品信息，生成回应用户需求且用户容易理解的通俗回复。 | According to the information obtained in the pre-sales dialogue in Fashion category, the attributes that guide the user's needs, and the product information that meets the user's needs, generate a popular reply that responds to the user's needs and is easy for the user to understand. |
| Input | 售前对话：用户：还有没有其他款推荐 已获取的用户需求偏好信息：款式： 引导用户需求的属性：材质： | Pre-sale dialogue: User: Do you have any other recommendations? Obtained user demand preference information: Style: Attributes that guide user demand: Material: |
| Output | 要什么材质的呢亲爱哒 | What material do you want dear? |