# OpenReview forum: "Conversational Recommender System and Large Language Model Are Made for Each Other in E-commerce Pre-sales Dialogue"
_EMNLP/2023/Conference — EMNLP 2023 Findings_

### Official Review · Reviewer_n8cG · 2023-08-03

**Typos Grammar Style And Presentation Improvements:** 1. In Table 4, Dist-1 for Chinese-Alp…
**Soundness:** 3

**Excitement:**

2: Mediocre: This paper makes marginal contributions (vs non-contemporaneous work), so I would rather not see it in the conference.

**Missing References:**

As described in Reasons to Reject:
"What does BERT know about books, movies and music? Probing BERT for conversational recommendation" (Penha et al., RecSys 2020);
"'It doesn't look good for a date': Transforming Critiques into Preferences for Conversational Recommendation Systems" (Bursztyn et al., EMNLP 2021);
"ReXPlug: Explainable Recommendation using Plug-and-Play Language Model" (Hada and Shevade, SIGIR 2021).


**Paper Topic And Main Contributions:**

This paper focuses on the combination of LLMs and CRSs in the context of product recommendations. Authors approach the topic from two viewpoints --- either "LLMs assisting CRSs", or "CRSs assisting LLMs" --- and across four sub-tasks implied in recommendation-making: intent understanding, preference elicitation, product recommendation, and response generation. Evaluation measures depend on the sub-task: precision, recall, and F1 for intent understanding and preference elicitation; accuracy, hit@5 and MRR@5 for product recommendation; and distinct@1, informativeness, and relevance for response generation.
Authors rely on the U-NEED dataset, which comprises 1135-1748 dialogues organized across five product categories: Beauty, Phones, Fashion, Shoes, and Eletronics. More simple baselines vary for each sub-task, but more complex baselines generally include both CRSs (variations of UniMIND) and LLMs (ChatGLM-6B and Chinese-Alpaca) in isolation, as well as the proposed methods: LLMs combined into CRSs (ChatGLM/Alpaca combined into UniMIND), or CRSs combined into LLMs (UniMIND combined into ChatGLM/Alpaca). Findings are somewhat unclear when comparing the proposed methods against the more complex baselines, with marginal gains associated to either "LLMs assisting CRSs" or "CRSs assisting LLMs" depending on the sub-task.

**Questions For The Authors:**

1. Given the observations in Reasons to Reject (#1), how do you view the difference in model capacity between the proposed methods and the baselines?
2. Lines 473-474 claim that "CRSs assisting LLMs" can improve the "robustness" of the LLMs alone. Are CRSs improving LLMs' robustness or are they improving their domain knowledge?


**Reasons To Accept:**

1. The paper focuses on a relevant topic with direct, real-world applicability.
2. Experiments contemplate five product categories, and include sub-tasks that are consensually important to conversational recommendations. The product recommendation sub-task has received increasing attention from the NLP community, particularly in how to leverage LLMs for the task.
3. Interestingly, it is for the product recommendation sub-task that authors report the most promising gains (as seen in aggregated accuracy of ALLM-CCRS, Table 3). A paper focused on "LLMs assisting CRSs" applied to the product recommendation sub-task could lead to potentially helpful findings to the community.

**Reasons To Reject:**

1. Since there are smaller LMs behind UniMIND (e.g., BART) and larger LMs representing LLMs, the experimental set up makes it hard to understand if the gains are due to the proposed methods or simply due to increased model capacity. Comparing the combination of LLMs + CRSs to the LLMs alone shows small gains (e.g., on Tables 1 and 4), so one conclusion can be that these marginal gains are due to the additional parameters introduced by the CRS --- in this case, would a slightly larger LLM be enough to achieve similar gains?
2. Methodology aside, I am not sure how novel and practically helpful the findings are. Essentially, the main takeaway is that it depends on the nature of the sub-task how to combine LLMs and CRSs. This takeaway has already been contributed by various previous works, for example:
a) "What does BERT know about books, movies and music? Probing BERT for conversational recommendation" (Penha et al., 2020) shows how LLMs can be helpful in recommendation-making;
b) "'It doesn't look good for a date': Transforming Critiques into Preferences for Conversational Recommendation Systems" (Bursztyn et al., 2021) shows how LLMs can be helpful in preference interpretation;
c) "ReXPlug: Explainable Recommendation using Plug-and-Play Language Model" (Hada and Shevade, 2021) shows how LLMs can be helpful in recommendation explanation.
However, these previous contributions are not properly acknowledged in the paper.
3. The paper makes too big of a claim in that it is the first to investigate the combination of LLMs + CRSs (e.g., line 14, lines 97-100). Lines 135-137 are not accurate (see #2 above).
4. The majority of the Appendices is copied *verbatim* from "U-NEED: A Fine-grained Dataset for User Needs-Centric E-Commerce Conversational Recommendation" (Liu et al., 2023), even duplicated in lines 979-994.

**Reproducibility:**

3: Could reproduce the results with some difficulty. The settings of parameters are underspecified or subjectively determined; the training/evaluation data are not widely available.

**Reviewer Confidence:**

4: Quite sure. I tried to check the important points carefully. It's unlikely, though conceivable, that I missed something that should affect my ratings.

---

> ### Author Rebuttal · Authors · 2023-08-29
>
> ### Thank you for your careful review. We really appreciate your efforts in reviewing our paper.
>
> Our work provides many interesting experimental findings, mainly including but not limited to:
> 1. Main finding of our work：CRSs and LLMs can benefit each other.
> On the four tasks of Conversational Recommendation, the state-of-the-art experimental results of the collaborative approach compared with the non-collaborative approach always appear in the collaborative approach, and this phenomenon strongly proves the effectiveness of improving the performance of the conversational recommendation system through collaboration between the traditional CRS approach and the LLMs approach.
> 2. Effectiveness of collaborations varies across domains.
> For the results in the Shoes and Phones domains in Tasks 3 and 4, some of the metrics appear state-of-the-art in the non-collaborative approach. This indicates that our proposed collaborative methods have different effectiveness for different distribution data. The reason for this may be that the corresponding conversations under the Shoes and Phones domains are more homogeneous and the product information is scarce, so it is difficult to highlight the advantages of the model in utilizing the information under the collaborative approach. This suggests that the richness of information under the corresponding scenarios of the data needs to be evaluated beforehand when applying our proposed collaborative approach.
> 3. Effectiveness of collaborations varies across tasks.
> For the two different collaboration approaches, XLLM-YCRS and YCRS-XLLM, the strengths and limitations of the approaches are observed based on the experimental results of Task 1, 2 and 3.
> From Table 1-3, we observe that (i) XLLM-YCRS > YCRS-XLLM on Task 1 and (ii) XLLM-YCRS < YCRS-XLLM on Tasks 2 and 3. The experimental results show consistency with the model results under the non-collaborative approach (CRS > LLM on Task 1; CRS < LLM on Tasks 2 and 3). That is, when the better model is used as the subject and the other type of model is used for collaborative assistance, an overall improved result will be obtained.
> 4. Main finding on Task 1: LLMs show superb performance in understanding user needs.
> Table 1 shows that the LLMs approach achieves a performance that exceeds that of the CRSs and the classical approach. Moreover, the performance of LLMs is further improved by considering the prediction results of CRSs. We attribute this to the strong capability of the LLMs for dialogue understanding. Note that traditional CRS methods struggle to model the fine-grained needs and preferences in a dialogue. The emergence of LLMs, on the other hand, makes it possible for a pre-sales shopping robot to truly understand users' needs and preferences. In practice,  "CRS assisting LLM" approach may be a viable solution based on the results in Table 1.
> 5. Main finding on Task 2: CRSs and LLMs show limited performance in user needs elicitation.
> The capacity of LLMs to provide decision-making information in certain ways (e.g., chain of thoughts) is a key point of exploration in LLMs. In this work, we explore whether LLMs can select attributes that can elicit users' preferences. The results in Table 2 show the state-of-the-art performance of LLMs in assisting CRSs on the Precision metric. However, in the Recall metric it is the classical approach applied in industry that achieves state-of-the-art performance. Note that ChatGLM assisting CRSs improves performance, while Chinese-Alpaca assisting CRSs decreases performance. This may be an accumulation of errors due to the limited performance of Chinese-Alpaca on Task 2. The collaboration between LLMs and CRSs in terms of strategies leaves a lot of room for exploration. We plan to explore more sophisticated ways of collaboration in future work, such as LLMs and CRSs engaging in debates to decide which attributes should be chosen at the moment.
> 6. Main finding on Task 3:  LLMs (i.e. generative recommendation) can boost CRSs to achieve the best performance.
> Note that although we fine-tuned the LLMs with training data, the LLMs providing product recommendations is not a classification task. LLMs select and recommend product from a given candidate list based on understanding the given instruction and input. CRSs, on the other hand, model the product recommendation as a classification task, and its prediction is computed based on the relations between user representation and candidate product representation. We consider these to be two different approaches to providing recommendations. The results in Table 3 show that modeling the relations between user representations and candidate item representations is still effective, and the two ways of providing recommendations are mutually helpful. In detail, LLMs can enhance CRSs to achieve state-of-the-art performance, while CRSs can also help LLMs to get moderate performance. We believe that this is inspiring to study generative recommender systems for E-commerce scenarios.
> 7. Main finding on Task 4: CRSs and LLMs show comparable performance in pre-sales dialogue generation.
> It is widely recognized that the LLMs are much better at generating responses than the smaller models. However, Table 4 demonstrates that the performance of the two is comparable and does not exceed the human standard response. In the human evaluation: a score of 4 (or 5) means that a model-generated response meets (or exceeds) the ground truth response, respectively. And there is no major improvement in collaboration between the two. This means that there is still some room for improvement in pre-sales dialog generation. And the small model is still worth exploring because it can achieve comparable performance without spending a lot of resources and time for both training and inference.
>
> ### Thank you, Reviewer n8cG. Here are the responses to your concerns/questions.
>
> > these marginal gains are due to the additional parameters introduced by the CRS
>
> A: We do not think that collaboration "increases" the number of parameters in LLMs. We think this collaboration "considers" results from other perspectives. CRSs learn user, item, and attribute representations from E-commerce structured information to compute recommendation probabilities. While LLMs generate recommended products directly based on the understanding of the given instruction and input.
>
> > --- in this case, would a slightly larger LLM be enough to achieve similar gains?
>
> A: There might be. But that's not the same contribution as this paper.
>
> > Methodology aside, I am not sure how novel and practically helpful the findings are.
>
> A: We list the main findings above.
>
> > This takeaway has already been contributed by various previous works, ... for example: a) "What does BERT know about books, movies and music? Probing BERT for conversational recommendation" (Penha et al., 2020) shows how LLMs can be helpful in recommendation-making;
>
> A: BERT is a pre-trained language model, but not a large language model in terms of parameter scales.
>
> > However, these previous contributions are not properly acknowledged in the paper.
>
> A: We think that none of the language models (with ~100 million parameters) used in the literature listed may be viewed as LLMs (with at least 1 billion parameters).
>
> > The paper makes too big of a claim in that it is the first to investigate the combination of LLMs + CRSs (e.g., line 14, lines 97-100). Lines 135-137 are not accurate (see #2 above).
>
> A: We indicate the scope of this paper's contribution, E-commerce pre-sales dialogue, both in the title and in the end part of the Introduction. Thank you for your feedback. We will revise the statements in the corresponding parts to avoid ambiguity.
>
> > The majority of the Appendices is copied verbatim from "U-NEED: A Fine-grained Dataset for User Needs-Centric E-Commerce Conversational Recommendation" (Liu et al., 2023), even duplicated in lines 979-994.
>
> A: We apologize for this mistake. We will rewrite the experimental settings in the final version. It will not affect the experimental results and findings.
>
> > Given the observations in Reasons to Reject (#1), how do you view the difference in model capacity between the proposed methods and the baselines?
>
> A: In terms of the number of parameters, they may be incomparable. The number of LLMs parameters may be far more than the classical approach. But for a specific task, the difference in their performance can provide some insights into applying LLMs in real-world scenarios.
>
> > Lines 473-474 claim that "CRSs assisting LLMs" can improve the "robustness" of the LLMs alone. Are CRSs improving LLMs' robustness or are they improving their domain knowledge?
>
> A: Sorry for the ambiguity. The collaborative approach of this paper does not augment the domain knowledge of LLMs. From our point of view, it is difficult to make LLMs memorize a large amount of domain knowledge. A recent research hotspot is to enhance the domain knowledge of LLMs by means of retrieval. However, in this way, LLMs still need to process the information retrieved back. We would like to have small models like CRSs that can learn the structured information of real scenarios to process some information on complex relationships. And the LLMs can directly utilize this information in a collaborative way.

---

### Official Review · Reviewer_FfbT · 2023-08-05

**Soundness:** 3

**Excitement:**

2: Mediocre: This paper makes marginal contributions (vs non-contemporaneous work), so I would rather not see it in the conference.

**Paper Topic And Main Contributions:**

The paper explores the collaboration of conversational recommender systems (CRSs) and large language models (LLMs) in E-commerce pre-sales dialogues. The paper proposes two methods of collaboration: CRS assisting LLM and LLM assisting CRS. The paper evaluates the effectiveness of these methods on a real-world dataset of pre-sales dialogues, involving four tasks: dialogue understanding, needs elicitation, recommendation, and dialogue generation. The paper shows that the collaboration of CRS and LLM can improve the performance of these tasks.

**Reasons To Accept:**

1. Using LLMs to assist in recommendation tasks is an interesting problem.
2. The authors' proposal to combine LLMs with traditional CRS models is well-motivated.

**Reasons To Reject:**

1. The paper's writing needs improvement. Four different tasks are defined in the paper, but the details of each task are not clearly explained. For specifics, see the Typos, Grammar, Style, and Presentation Improvements section.
2. The proposed method only achieves improvements on some specific CRS tasks.
3. There is a lack of explanation for instances where no improvement is observed. For example, in line 406, "In contrast, the performance in task 1 across all categories declines."
4. The authors only conducted experiments on a single dataset, which could be a limitation.
5. No analysis of computational overhead is provided.

**Reproducibility:**

4: Could mostly reproduce the results, but there may be some variation because of sample variance or minor variations in their interpretation of the protocol or method.

**Reviewer Confidence:**

3: Pretty sure, but there's a chance I missed something. Although I have a good feel for this area in general, I did not carefully check the paper's details, e.g., the math, experimental design, or novelty.

**Typos Grammar Style And Presentation Improvements:**

The paper's writing and the content of the corresponding figures need revision to make them easier to understand. For example,

1. Section 3.3 does not introduce each task in multi-task training, and there is no specific explanation or example for the design of prompts. Figure 3 and the Figures in the Appendix are also difficult to understand.
2. The variable name definitions in Section 3.3 are problematic. $X_S$ is defined but not used. $X_R$ and $Z_R$ are not defined. The user needs-based recommendation task corresponding to Eq. 2 and Eq. 3 is not defined. Section 3.5 has similar issues.
3. In the experimental analysis section, the author uses tasks 1, 2, 3, and 4 to represent the tasks. Using task in Table 1, 2, 3, 4 or directly using the task name are more suitable.

---

> ### Author Rebuttal · Authors · 2023-08-29
>
> ### Thank you for your careful review. We really appreciate your efforts in reviewing our paper.
> Large language models(LLMs) have become a hot topic in AI research. On most tasks, LLMs can easily achieve performance far beyond that of the state-of-the-art methods with only a small amount of fine-tune data. Many researchers have begun to explore the capabilities and limitations of LLMs models for various tasks and scenarios.
>
> We would like to emphasize that our work can provide valuable insights for the future research and application of LLMs in E-commerce scenarios.
> 1. We explore the capacities of LLMs in real-world E-commerce scenarios, i.e. E-commerce pre-sales dialogues.
> Online shopping has become an inaccessible part of life. Pre-sales dialogue is a key factor in improving user experience and increasing purchase rates. The dataset we use is constructed based on real conversations between users and customer service staff, rather than conversations simulated by crowdsourcers. Our work explores four important tasks in E-commerce pre-sales dialogue that can provide insights for applying LLMs in e-commerce scenarios as well as developing LLMs-based customer service assistants.
> 2. We explore collaborations between LLMs and conversational recommender systems(CRSs).
> LLMs generate answers based on massive pre-trained knowledge and understanding of the given instruction and input. CRSs, on the other hand, learn representations of users, items, and attributes to compute probabilities. We view these as two different approaches to solving questions. Our work reports the results of combining these two types of approaches, which is instructive for subsequent research (open-domain LLMs base + small models with domain-specific knowledge).
> 3. We conduct extensive experiments that show many interesting findings.
> We explore 2 types of collaboration on 4 important tasks, 4 product categories, 2 LLMs and 2 CRSs.
> Based on the experimental results, we can analyze the differences and impacts of (1) tasks (2) collaboration methods (3) base models (4) product categories. See below for details.
>
> Our work provides many interesting experimental findings, mainly including but not limited to:
> 1. Main finding of our work：CRSs and LLMs can benefit each other.
> On the four tasks of Conversational Recommendation, the state-of-the-art experimental results of the collaborative approach compared with the non-collaborative approach always appear in the collaborative approach, and this phenomenon strongly proves the effectiveness of improving the performance of the conversational recommendation system through collaboration between the traditional CRS approach and the LLMs approach.
> 2. Effectiveness of collaborations varies across domains.
> For the results in the Shoes and Phones domains in Tasks 3 and 4, some of the metrics appear state-of-the-art in the non-collaborative approach. This indicates that our proposed collaborative methods have different effectiveness for different distribution data. The reason for this may be that the corresponding conversations under the Shoes and Phones domains are more homogeneous and the product information is scarce, so it is difficult to highlight the advantages of the model in utilizing the information under the collaborative approach. This suggests that the richness of information under the corresponding scenarios of the data needs to be evaluated beforehand when applying our proposed collaborative approach.
> 3. Effectiveness of collaborations varies across tasks.
> For the two different collaboration approaches, XLLM-YCRS and YCRS-XLLM, the strengths and limitations of the approaches are observed based on the experimental results of Task 1, 2 and 3.
> From Table 1-3, we observe that (i) XLLM-YCRS > YCRS-XLLM on Task 1 and (ii) XLLM-YCRS < YCRS-XLLM on Tasks 2 and 3. The experimental results show consistency with the model results under the non-collaborative approach (CRS > LLM on Task 1; CRS < LLM on Tasks 2 and 3). That is, when the better model is used as the subject and the other type of model is used for collaborative assistance, an overall improved result will be obtained.
> 4. Main finding on Task 1: LLMs show superb performance in understanding user needs.
> Table 1 shows that the LLMs approach achieves a performance that exceeds that of the CRSs and the classical approach. Moreover, the performance of LLMs is further improved by considering the prediction results of CRSs. We attribute this to the strong capability of the LLMs for dialogue understanding. Note that traditional CRS methods struggle to model the fine-grained needs and preferences in a dialogue. The emergence of LLMs, on the other hand, makes it possible for a pre-sales shopping robot to truly understand users' needs and preferences. In practice,  "CRS assisting LLM" approach may be a viable solution based on the results in Table 1.
> 5. Main finding on Task 2: CRSs and LLMs show limited performance in user needs elicitation.
> The capacity of LLMs to provide decision-making information in certain ways (e.g., chain of thoughts) is a key point of exploration in LLMs. In this work, we explore whether LLMs can select attributes that can elicit users' preferences. The results in Table 2 show the state-of-the-art performance of LLMs in assisting CRSs on the Precision metric. However, in the Recall metric it is the classical approach applied in industry that achieves state-of-the-art performance. Note that ChatGLM assisting CRSs improves performance, while Chinese-Alpaca assisting CRSs decreases performance. This may be an accumulation of errors due to the limited performance of Chinese-Alpaca on Task 2. The collaboration between LLMs and CRSs in terms of strategies leaves a lot of room for exploration. We plan to explore more sophisticated ways of collaboration in future work, such as LLMs and CRSs engaging in debates to decide which attributes should be chosen at the moment.
> 6. Main finding on Task 3:  LLMs (i.e. generative recommendation) can boost CRSs to achieve the best performance.
> Note that although we fine-tuned the LLMs with training data, the LLMs providing product recommendations is not a classification task. LLMs select and recommend product from a given candidate list based on understanding the given instruction and input. CRSs, on the other hand, model the product recommendation as a classification task, and its prediction is computed based on the relations between user representation and candidate product representation. We consider these to be two different approaches to providing recommendations. The results in Table 3 show that modeling the relations between user representations and candidate item representations is still effective, and the two ways of providing recommendations are mutually helpful. In detail, LLMs can enhance CRSs to achieve state-of-the-art performance, while CRSs can also help LLMs to get moderate performance. We believe that this is inspiring to study generative recommender systems for E-commerce scenarios.
> 7. Main finding on Task 4: CRSs and LLMs show comparable performance in pre-sales dialogue generation.
> It is widely recognized that the LLMs are much better at generating responses than the smaller models. However, Table 4 demonstrates that the performance of the two is comparable and does not exceed the human standard response. In the human evaluation: a score of 4 (or 5) means that a model-generated response meets (or exceeds) the ground truth response, respectively. And there is no major improvement in collaboration between the two. This means that there is still some room for improvement in pre-sales dialog generation. And the small model is still worth exploring because it can achieve comparable performance without spending a lot of resources and time for both training and inference.
>
>
> ### Thank you, Reviewer FfbT. Here are the responses to your concerns/questions.
>
> > The paper's writing needs improvement. Four different tasks are defined in the paper, but the details of each task are not clearly explained. For specifics, see the Typos, Grammar, Style, and Presentation Improvements section.
>
> A: We appreciate your time and constructive review. We will revise the writing of this paper following your suggestions. Also, we will rewrite the Results and Analysis section to include the findings we listed above. We will also tweak other sections, such as the Appendix, to make this paper easy to understand.
>
> > The proposed method only achieves improvements on some specific CRS tasks.
>
> A: We list the main findings of the experimental results above.
>
> > There is a lack of explanation for instances where no improvement is observed. For example, in line 406, "In contrast, the performance in task 1 across all categories declines."
>
> A: Thanks for your suggestion, we will rewrite the analysis of experimental results in the final version and add more explanations.
>
> > The authors only conducted experiments on a single dataset, which could be a limitation.
>
> A: To the best of our knowledge, there are no other accessible datasets in the field of E-commerce pre-sales dialogue.  The dataset we use contains five product categories, i.e. Beauty, Phones, Fashion, Shoes and Electronics, which somewhat alleviates this limitation.
>
> > No analysis of computational overhead is provided.
>
> A: This is due to the limitation of the length of the submission. For CRSs, we use a NVIDIA A100-SXM4-80GB gpu and train model for 10 epochs, with a duration of approximately 12 hours. For LLMs, we use a NVIDIA A100-SXM4-80GB gpu and train model for 3 epochs, with a duration of approximately 9 hours. Thank you for your suggestion and we will add implementation details in the final version.

---

### Official Review · Reviewer_MtkJ · 2023-08-10

**Soundness:** 3

**Excitement:**

3: Ambivalent: It has merits (e.g., it reports state-of-the-art results, the idea is nice), but there are key weaknesses (e.g., it describes incremental work), and it can significantly benefit from another round of revision. However, I won't object to accepting it if my co-reviewers champion it.

**Missing References:**

None.

**Paper Topic And Main Contributions:**

This work proposes to combine a large language model (LLM) and a conversational recommender system (CRS) for precise recommendation. Experiments show the effectiveness of this kind of collaboration.

**Questions For The Authors:**

Please refer to **Reasons To Reject**.

**Reasons To Accept:**

1. The authors explore a novel setting in conversational recommendation, i.e., LLM collaborating with CRS.

**Reasons To Reject:**

1. The motivation for using CRSs for recommendation is not clear. The authors should justify why it is not sufficient to fine-tune an LLM on the recommendation task. The setting **no collaboration** in Table 1,2,3,4 shows that the proposed method has only a marginal advantage over the single LLM, despite the incorporation of the collaboration mechanism.
2. The combination of the CRSs and the LLM is kind of simple and ad hoc. The authors should explain whether there are any alternatives or ablations to the proposed interaction mechanism.
3. The term **task 3** in Line 401 is undefined and confusing. The authors should clarify what it refers to and how it relates to the previous tasks or sections of the paper.
4. It might be inappropriate to move part of the analysis of the **main** result into the appendix. The authors should include the most important findings and insights in the main paper. Moreover, the figure/table referred in Appendix A is missing or mislabeled.
5. The writing of the experiment section is messy and hard to follow. The authors should improve the clarity and coherence of their presentation, and avoid grammatical errors and typos.

**Reproducibility:**

4: Could mostly reproduce the results, but there may be some variation because of sample variance or minor variations in their interpretation of the protocol or method.

**Reviewer Confidence:**

3: Pretty sure, but there's a chance I missed something. Although I have a good feel for this area in general, I did not carefully check the paper's details, e.g., the math, experimental design, or novelty.

---

> ### Author Rebuttal · Authors · 2023-08-29
>
> ### Thank you for your careful review. We really appreciate your efforts in reviewing our paper.
> Large language models(LLMs) have become a hot topic in AI research. On most tasks, LLMs can easily achieve performance far beyond that of the state-of-the-art methods with only a small amount of fine-tune data. Many researchers have begun to explore the capabilities and limitations of LLMs models for various tasks and scenarios.
>
> We would like to emphasize that our work can provide valuable insights for the future research and application of LLMs in E-commerce scenarios.
> 1. We explore the capacities of LLMs in real-world E-commerce scenarios, i.e. E-commerce pre-sales dialogues.
> Online shopping has become an inaccessible part of life. Pre-sales dialogue is a key factor in improving user experience and increasing purchase rates. The dataset we use is constructed based on real conversations between users and customer service staff, rather than conversations simulated by crowdsourcers. Our work explores four important tasks in E-commerce pre-sales dialogue that can provide insights for applying LLMs in e-commerce scenarios as well as developing LLMs-based customer service assistants.
> 2. We explore collaborations between LLMs and conversational recommender systems(CRSs).
> LLMs generate answers based on massive pre-trained knowledge and understanding of the given instruction and input. CRSs, on the other hand, learn representations of users, items, and attributes to compute probabilities. We view these as two different approaches to solving questions. Our work reports the results of combining these two types of approaches, which is instructive for subsequent research (open-domain LLMs base + small models with domain-specific knowledge).
> 3. We conduct extensive experiments that show many interesting findings.
> We explore 2 types of collaboration on 4 important tasks, 4 product categories, 2 LLMs and 2 CRSs.
> Based on the experimental results, we can analyze the differences and impacts of (1) tasks (2) collaboration methods (3) base models (4) product categories. See below for details.
>
> Our work provides many interesting experimental findings, mainly including but not limited to:
> 1. Main finding of our work：CRSs and LLMs can benefit each other.
> On the four tasks of Conversational Recommendation, the state-of-the-art experimental results of the collaborative approach compared with the non-collaborative approach always appear in the collaborative approach, and this phenomenon strongly proves the effectiveness of improving the performance of the conversational recommendation system through collaboration between the traditional CRS approach and the LLMs approach.
> 2. Effectiveness of collaborations varies across domains.
> For the results in the Shoes and Phones domains in Tasks 3 and 4, some of the metrics appear state-of-the-art in the non-collaborative approach. This indicates that our proposed collaborative methods have different effectiveness for different distribution data. The reason for this may be that the corresponding conversations under the Shoes and Phones domains are more homogeneous and the product information is scarce, so it is difficult to highlight the advantages of the model in utilizing the information under the collaborative approach. This suggests that the richness of information under the corresponding scenarios of the data needs to be evaluated beforehand when applying our proposed collaborative approach.
> 3. Effectiveness of collaborations varies across tasks.
> For the two different collaboration approaches, XLLM-YCRS and YCRS-XLLM, the strengths and limitations of the approaches are observed based on the experimental results of Task 1, 2 and 3.
> From Table 1-3, we observe that (i) XLLM-YCRS > YCRS-XLLM on Task 1 and (ii) XLLM-YCRS < YCRS-XLLM on Tasks 2 and 3. The experimental results show consistency with the model results under the non-collaborative approach (CRS > LLM on Task 1; CRS < LLM on Tasks 2 and 3). That is, when the better model is used as the subject and the other type of model is used for collaborative assistance, an overall improved result will be obtained.
> 4. Main finding on Task 1: LLMs show superb performance in understanding user needs.
> Table 1 shows that the LLMs approach achieves a performance that exceeds that of the CRSs and the classical approach. Moreover, the performance of LLMs is further improved by considering the prediction results of CRSs. We attribute this to the strong capability of the LLMs for dialogue understanding. Note that traditional CRS methods struggle to model the fine-grained needs and preferences in a dialogue. The emergence of LLMs, on the other hand, makes it possible for a pre-sales shopping robot to truly understand users' needs and preferences. In practice,  "CRS assisting LLM" approach may be a viable solution based on the results in Table 1.
> 5. Main finding on Task 2: CRSs and LLMs show limited performance in user needs elicitation.
> The capacity of LLMs to provide decision-making information in certain ways (e.g., chain of thoughts) is a key point of exploration in LLMs. In this work, we explore whether LLMs can select attributes that can elicit users' preferences. The results in Table 2 show the state-of-the-art performance of LLMs in assisting CRSs on the Precision metric. However, in the Recall metric it is the classical approach applied in industry that achieves state-of-the-art performance. Note that ChatGLM assisting CRSs improves performance, while Chinese-Alpaca assisting CRSs decreases performance. This may be an accumulation of errors due to the limited performance of Chinese-Alpaca on Task 2. The collaboration between LLMs and CRSs in terms of strategies leaves a lot of room for exploration. We plan to explore more sophisticated ways of collaboration in future work, such as LLMs and CRSs engaging in debates to decide which attributes should be chosen at the moment.
> 6. Main finding on Task 3:  LLMs (i.e. generative recommendation) can boost CRSs to achieve the best performance.
> Note that although we fine-tuned the LLMs with training data, the LLMs providing product recommendations is not a classification task. LLMs select and recommend product from a given candidate list based on understanding the given instruction and input. CRSs, on the other hand, model the product recommendation as a classification task, and its prediction is computed based on the relations between user representation and candidate product representation. We consider these to be two different approaches to providing recommendations. The results in Table 3 show that modeling the relations between user representations and candidate item representations is still effective, and the two ways of providing recommendations are mutually helpful. In detail, LLMs can enhance CRSs to achieve state-of-the-art performance, while CRSs can also help LLMs to get moderate performance. We believe that this is inspiring to study generative recommender systems for E-commerce scenarios.
> 7. Main finding on Task 4: CRSs and LLMs show comparable performance in pre-sales dialogue generation.
> It is widely recognized that the LLMs are much better at generating responses than the smaller models. However, Table 4 demonstrates that the performance of the two is comparable and does not exceed the human standard response. In the human evaluation: a score of 4 (or 5) means that a model-generated response meets (or exceeds) the ground truth response, respectively. And there is no major improvement in collaboration between the two. This means that there is still some room for improvement in pre-sales dialog generation. And the small model is still worth exploring because it can achieve comparable performance without spending a lot of resources and time for both training and inference.
>
> ### Thank you, Reviewer MtkJ. Here are the responses to your concerns/questions.
>
> > The motivation for using CRSs for recommendation is not clear.
>
> A: CRSs can do all the tasks of E-commerce pre-sales dialogue. We guess you might be asking why we chose CRSs to collaborate with LLMs. We focus on exploring the capabilities of LLMs for E-commerce pre-sales dialogue, which is a specific scenario of conversational recommendation.
> CRSs learn user, item, and attribute representations to compute recommendation probabilities. While LLMs generate recommended products directly based on the understanding of the given instruction and input.
> We focus on exploring the performance of the combination of these two types of approaches for four important tasks in E-commerce pre-sales dialogue.
>
> > The authors should justify why it is not sufficient to fine-tune an LLM on the recommendation task.
>
> A: First the performance of LLMs on user needs-based recommendation is relatively limited. We think that the possible reason is that LLMs do not have the product-related knowledge. Although fine-tuning methods enable LLMs to be more familiar with the instructions and inputs of the tasks, it is still difficult for LLMs to retain the knowledge of the related attributes of the products. In addition, in real scenarios, the number of related candidate products may be more than 50,000 and the products are updated frequently, fine-tuning is not feasible as it consumes a lot of resources.
>
> > The setting of no collaboration in Table 1,2,3,4 shows that the proposed method has only a marginal advantage over the single LLM, despite the incorporation of the collaboration mechanism.
>
> A: In Table 3, on all five categories, ALLM-CCRS shows a considerable improvement in Acc (0.2639->0.2822), H@5 (0.5617->0.5832), and M@5 (0.3737->0.3919) compared to the CRS method without collaboration. In addition, we list the main findings of the experimental results above.
>
> > The combination of the CRSs and the LLM is kind of simple and ad hoc. The authors should explain whether there are any alternatives or ablations to the proposed interaction mechanism.
>
> A: The collaboration approach we used can be seen as a simplified version of a mixture of experts. As we state in the response to the first question, LLMs and CRSs have different approaches to completing tasks. And we have considered other ways for LLMs to collaborate with CRS, such as joint training. However the time and GPU resources required by fine-tuing LLMs don't really allow LLMs to have complex interactions with CRSs. Thank you for your advice. We will explore more ways for LLMs to collaborate with smaller models based on the findings of this work.
>
> > The term task 3 in Line 401 is undefined and confusing. The authors should clarify what it refers to and how it relates to the previous tasks or sections of the paper.
>
> A: Task 3 refers to user needs-based recommendation task. The corresponding experimental results are in Table 3.
>
> > It might be inappropriate to move part of the analysis of the main result into the appendix. The authors should include the most important findings and insights in the main paper. Moreover, the figure/table referred in Appendix A is missing or mislabeled. The writing of the experiment section is messy and hard to follow. The authors should improve the clarity and coherence of their presentation, and avoid grammatical errors and typos.
>
> A: Thanks for pointing out this problem. We appreciate your time. We have reorganized the findings of the experimental results. We list the main findings above. We will rewrite the section of experimental results analysis in the final version.

---

### Official Review · Reviewer_j99w · 2023-08-10

**Typos Grammar Style And Presentation Improvements:** N.A.
**Soundness:** 4

**Excitement:**

4: Strong: This paper deepens the understanding of some phenomenon or lowers the barriers to an existing research direction.

**Missing References:**

N.A.

**Paper Topic And Main Contributions:**

In this paper, the authors study the effectiveness of combing large language model with conversational recommendation system in e-commerce pre-sales dialogues. They propose two collaboration methods, i.e., conversational recommender systems assisting large language model and large language model assisting conversational recommender systems. The authors have performed extensive experiments on real datasets to demonstrate the effectiveness of the proposed methods.

**Questions For The Authors:**

1. What are the specific properties of e-commerce pre-sale dialogues have been explored by the proposed model?

2. Why using different evaluation metrics in Table 1, 2, and 3?

**Reasons To Accept:**

1. The authors propose to study the combination of large language model and conversational recommendation system in pre-sales dialogue in e-commerce. This idea seems novel.

2. The authors propose two collaboration methods: 1) conversational recommendation system assisting large language model, and 2) large language model assisting conversational recommendation system.

3. The authors have performed extensive experiments on a real-world pre-sale dialogue dataset in e-commerce.

4. This paper is well-structured and clearly written.

**Reasons To Reject:**

1. The main objective of this work is to explore the integration of conversational recommendation system with large language model in e-commerce pre-sale dialogues. However, in the proposed models, the specific properties of e-commerce pre-sales dialogues are not explored.

2. According to my understanding, the proposed two collaboration models can also be applied to other conversational recommendation scenarios, e.g., movie recommendation. However, in this work, the authors only study the performance of the proposed methods on one e-commerce pre-sale dialogue dataset. Thus, the experimental analysis is not sufficient.

**Reproducibility:**

3: Could reproduce the results with some difficulty. The settings of parameters are underspecified or subjectively determined; the training/evaluation data are not widely available.

**Reviewer Confidence:**

4: Quite sure. I tried to check the important points carefully. It's unlikely, though conceivable, that I missed something that should affect my ratings.

---

> ### Author Rebuttal · Authors · 2023-08-29
>
> ### Thank you for your careful review. We really appreciate your efforts in reviewing our paper.
>
> Our work provides many interesting experimental findings, mainly including but not limited to:
> 1. Main finding of our work：CRSs and LLMs can benefit each other.
> On the four tasks of Conversational Recommendation, the state-of-the-art experimental results of the collaborative approach compared with the non-collaborative approach always appear in the collaborative approach, and this phenomenon strongly proves the effectiveness of improving the performance of the conversational recommendation system through collaboration between the traditional CRS approach and the LLMs approach.
> 2. Effectiveness of collaborations varies across domains.
> For the results in the Shoes and Phones domains in Tasks 3 and 4, some of the metrics appear state-of-the-art in the non-collaborative approach. This indicates that our proposed collaborative methods have different effectiveness for different distribution data. The reason for this may be that the corresponding conversations under the Shoes and Phones domains are more homogeneous and the product information is scarce, so it is difficult to highlight the advantages of the model in utilizing the information under the collaborative approach. This suggests that the richness of information under the corresponding scenarios of the data needs to be evaluated beforehand when applying our proposed collaborative approach.
> 3. Effectiveness of collaborations varies across tasks.
> For the two different collaboration approaches, XLLM-YCRS and YCRS-XLLM, the strengths and limitations of the approaches are observed based on the experimental results of Task 1, 2 and 3.
> From Table 1-3, we observe that (i) XLLM-YCRS > YCRS-XLLM on Task 1 and (ii) XLLM-YCRS < YCRS-XLLM on Tasks 2 and 3. The experimental results show consistency with the model results under the non-collaborative approach (CRS > LLM on Task 1; CRS < LLM on Tasks 2 and 3). That is, when the better model is used as the subject and the other type of model is used for collaborative assistance, an overall improved result will be obtained.
> 4. Main finding on Task 1: LLMs show superb performance in understanding user needs.
> Table 1 shows that the LLMs approach achieves a performance that exceeds that of the CRSs and the classical approach. Moreover, the performance of LLMs is further improved by considering the prediction results of CRSs. We attribute this to the strong capability of the LLMs for dialogue understanding. Note that traditional CRS methods struggle to model the fine-grained needs and preferences in a dialogue. The emergence of LLMs, on the other hand, makes it possible for a pre-sales shopping robot to truly understand users' needs and preferences. In practice,  "CRS assisting LLM" approach may be a viable solution based on the results in Table 1.
> 5. Main finding on Task 2: CRSs and LLMs show limited performance in user needs elicitation.
> The capacity of LLMs to provide decision-making information in certain ways (e.g., chain of thoughts) is a key point of exploration in LLMs. In this work, we explore whether LLMs can select attributes that can elicit users' preferences. The results in Table 2 show the state-of-the-art performance of LLMs in assisting CRSs on the Precision metric. However, in the Recall metric it is the classical approach applied in industry that achieves state-of-the-art performance. Note that ChatGLM assisting CRSs improves performance, while Chinese-Alpaca assisting CRSs decreases performance. This may be an accumulation of errors due to the limited performance of Chinese-Alpaca on Task 2. The collaboration between LLMs and CRSs in terms of strategies leaves a lot of room for exploration. We plan to explore more sophisticated ways of collaboration in future work, such as LLMs and CRSs engaging in debates to decide which attributes should be chosen at the moment.
> 6. Main finding on Task 3:  LLMs (i.e. generative recommendation) can boost CRSs to achieve the best performance.
> Note that although we fine-tuned the LLMs with training data, the LLMs providing product recommendations is not a classification task. LLMs select and recommend product from a given candidate list based on understanding the given instruction and input. CRSs, on the other hand, model the product recommendation as a classification task, and its prediction is computed based on the relations between user representation and candidate product representation. We consider these to be two different approaches to providing recommendations. The results in Table 3 show that modeling the relations between user representations and candidate item representations is still effective, and the two ways of providing recommendations are mutually helpful. In detail, LLMs can enhance CRSs to achieve state-of-the-art performance, while CRSs can also help LLMs to get moderate performance. We believe that this is inspiring to study generative recommender systems for E-commerce scenarios.
> 7. Main finding on Task 4: CRSs and LLMs show comparable performance in pre-sales dialogue generation.
> It is widely recognized that the LLMs are much better at generating responses than the smaller models. However, Table 4 demonstrates that the performance of the two is comparable and does not exceed the human standard response. In the human evaluation: a score of 4 (or 5) means that a model-generated response meets (or exceeds) the ground truth response, respectively. And there is no major improvement in collaboration between the two. This means that there is still some room for improvement in pre-sales dialog generation. And the small model is still worth exploring because it can achieve comparable performance without spending a lot of resources and time for both training and inference.
>
> ### Thank you, Reviewer j99w. Here are the responses to your concerns/questions.
> > ... the proposed two collaboration models can also be applied to other conversational recommendation scenarios, e.g., movie recommendation.
>
> A: Yes, You're right. However, user needs-based recommendation is very different from movie recommendation. The LLMs has movie knowledge, so many studies have found that the LLMs can do well on the movie recommendation task even in the 0-shot case[1, 2]. We focus on product recommendation, which contains knowledge that LLMs do not have. From the experimental results, we also see that the LLMs has good performance on pre-sales dialog understanding (Table 1) and more limited performance on user needs-based recommendation (Table 3). Even after fine-tuning, it could not outperform the CRS SOTA approach. This shows that how to make LLMs model structured product attribute information is an important challenge for LLMs applied in e-commerce recommendation.
> This paper is trying to show the insight that CRS and LLMs are complementary on pre-sales dialogue tasks. This finding may be instructive for recommendations in other domains, but may not be entirely applicable.
> From our experience, it depends on the knowledge of ITEM. Movies, with a lot of textual information, movie summary, movie reviews, etc., are suitable for LLMs to "read". Products, on the other hand, relate to a lot of structured knowledge, such as user actions, product attributes, etc., which are not easily understood by LLMs. Moreover, when we transform them into instructions and inputs, there is also missing information.
>
> > ... only ... on one e-commerce pre-sale dialogue dataset. Thus, the experimental analysis is not sufficient.
>
> A:  For the exploration of the field of E-commerce pre-sales dialogue, we believe that our experiments are sufficient and provide a lot of interesting findings. Thank you for your suggestion, and in future work we will explore whether similar findings exist in other domains of recommendations.
>
> > What are the specific properties of e-commerce pre-sale dialogues have been explored by the proposed model?
>
> A：We list the main findings of our experimental results above, which can hopefully answer your question.
>
> > Why using different evaluation metrics in Table 1, 2, and 3?
>
> A: For the metrics in Tables 1 and 2, we adopt the benchmark and evaluation metrics proposed in the U-NEED dataset paper[3]. Regarding the user needs-based recommendation task, the evaluation metrics in [3] are Hit@10, Hit@50 and MRR@50. Due to the limitation of the input length of LLMs, where each product contains attributes and attribute values, we can provide a maximum of 20 candidate products. Therefore, in order to compare whether the collaborative approach improves the performance of CRSs, we measure Acc, Hit@5 and MRR@5.
>
> [1] Wang, Xiaolei, et al. "Rethinking the Evaluation for Conversational Recommendation in the Era of Large Language Models." arXiv preprint arXiv:2305.13112 (2023).
> [2]Hou, Yupeng, et al. "Large language models are zero-shot rankers for recommender systems." arXiv preprint arXiv:2305.08845 (2023).
> [3] Liu, Yuanxing, et al. "U-NEED: A Fine-grained Dataset for User Needs-Centric E-commerce Conversational Recommendation" . In Proceedings of the 46th International ACM SIGIR Conference on Research and Development in Information Retrieval (SIGIR '23). Association for Computing Machinery, New York, NY, USA, 2723–2732. https://doi.org/10.1145/3539618.3591878

---

### Meta-Review · Area_Chair_3tua · 2023-09-16

**Recommendation:** 3

**Metareview:**

This paper introduces approaches for combining the training of conversational recommender systems (CRS) and large language models (LLM) for pre-sales dialogue. They look at how CRS can assist LLM and vice versa. They look at four subtasks; intent understanding, preference elicitation, product recommendation, and response generation. They use the U-NEED dataset which has five product categories; beauty, phones, fashion, shoes, and electronics. The reviewers agree that this is a well motivated and interesting problem that has real world applications. The reviewers disagree on the clarity of the paper and have provided helpful suggestions for improvements. The authors provide detailed results and analysis that provide insights that future work may build off of. The reviewers range in excitement. The novelty is limited in comparison to recent works pointed out by the reviewers. Small improvements are seen in the results. Reviewer n8cG points out the potential confound in the number of model parameters that may contribute to the improvements. However, the work provides insights that will help inform future work in this area. The overclaim of novelty should be corrected in the next revision. Similarly, the plagiarized portion of the appendix must be rewritten and/or properly attributed in the next revision.

---

### Decision · Program_Chairs · 2023-10-07

**Decision:**

Accept-Findings

**Comment:**

This paper introduces approaches for combining the training of conversational recommender systems (CRS) and large language models (LLM) for pre-sales dialogue. They look at how CRS can assist LLM and vice versa. They look at four subtasks; intent understanding, preference elicitation, product recommendation, and response generation. They use the U-NEED dataset which has five product categories; beauty, phones, fashion, shoes, and electronics. The reviewers agree that this is a well motivated and interesting problem that has real world applications. The reviewers disagree on the clarity of the paper and have provided helpful suggestions for improvements. The authors provide detailed results and analysis that provide insights that future work may build off of. The reviewers range in excitement. The novelty is limited in comparison to recent works pointed out by the reviewers. Small improvements are seen in the results. Reviewer n8cG points out the potential confound in the number of model parameters that may contribute to the improvements. However, the work provides insights that will help inform future work in this area. The overclaim of novelty should be corrected in the next revision. Similarly, the plagiarized portion of the appendix must be rewritten and/or properly attributed in the next revision.